# STARE-VLA: Progressive Stage-Aware Reinforcement for Fine-Tuning Vision-Language-Action Models

## Abstract

Current Vision–Language–Action (VLA) models for robotic manipulation apply trajectory-level optimization (TPO/PPO) that suffers from coarse credit assignment, treating causally staged actions as uniform sequences. We introduce **St**age-**A**ware **Re**inforcement (**STARE**), a module that decomposes a long-horizon action trajectory into semantically meaningful stages and provides dense, interpretable, and stage-aligned reinforcement signals. Integrating STARE into TPO and PPO, we develop Stage-Aware TPO (**STA-TPO**) and Stage-Aware PPO (**STA-PPO**) for offline stage-wise preference and online intra-stage interaction, respectively. Further building on supervised fine-tuning as initialization, we propose the **I**mitation → **P**reference→ **I**nteraction (**IPI**), a serial fine-tuning pipeline for improving action accuracy in VLA models. Experiments on SimplerEnv and ManiSkill3 demonstrate substantial improvements, achieving state-of-the-art success rates of 98.0% and 96.4%, respectively, while exhibiting more stable and consistent training behavior. We further validate our approach on real-world robotic manipulation tasks. Our code will be released publicly.

## 1. Introduction

Large-scale Vision–Language–Action (VLA) models (Brohan et al., 2023; Zitkovich et al., 2023; Ghosh et al., 2024; Kim et al., 2024; Black et al., 2024; 2025) have recently emerged as powerful generalist policies for robotic manipulation. These models unify image, language, and action modalities within a single architecture, enabling robots to interpret multimodal inputs and generate executable actions.

Pretrained on massive-scale multimodal data (O'Neill et al., 2024; Walke et al., 2023), VLA models provide strong priors that can be efficiently adapted to diverse downstream tasks through fine-tuning, avoiding the need for retraining from scratch.

Current VLA models have been rapidly driven by the success of vision–language models (VLMs) and large language models (LLMs), as their output, i.e., action trajectories and sentences, can both be represented as sequential data (Zitkovich et al., 2023; O'Neill et al., 2024; Ghosh et al., 2024). Consequently, many developed fine-tuning techniques, such as supervised fine-tuning (SFT), Reinforcement Learning from Human Feedback (RLHF) (Ouyang et al., 2022), direct preference optimization (DPO) (Rafailov et al., 2023), Proximal Policy Optimization (PPO) (Schulman et al., 2017), and Group Relative Policy Optimization (GRPO) (Guo et al., 2025a), have been straightforwardly adopted for VLA models. However, fine-tuning on whole trajectories is often inefficient, as the large optimization space makes credit assignment across long-horizon trajectories highly ambiguous. Unlike language reasoning, where optimization depends on a holistic understanding of sentences without strict ordering, an action trajectory naturally decomposes into semantically distinct stages that are causally chained and vary in difficulty. For example, as in the pick-and-place task illustrated in Figure 1, *Reach* must precede *Grasp*, which in turn precedes *Transport* and *Place*. *Reach* and *Transport* are relatively easy with simple optimization objectives, while *Grasp* and *Place* are more challenging as they require precise geometric constraints. Overall task success hinges on correct progression through all stages. This fundamental characteristic motivates stage-aware objectives rather than monolithic trajectory-level optimization, which remains the predominant paradigm in current VLA fine-tuning.

In this paper, we design *Stage-Aware Reinforcement* (STARE), a plug-in module that decomposes action trajectories into progressive stages with dense reward signals based on task-specific semantics. Given a trajectory, either in the collected data or during the model's rollout, STARE employs a stage separator to identify *when* stage transitions occur, based on the translation and orientation of an

[1]Anonymous Institution, Anonymous City, Anonymous Region, Anonymous Country. Correspondence to: Anonymous Author <anon.email@domain.com>.

Preliminary work. Under review by the International Conference on Machine Learning (ICML). Do not distribute.

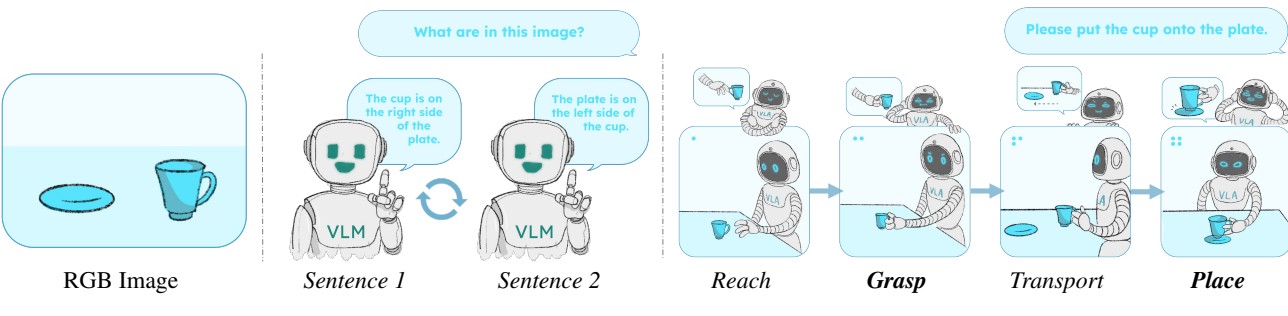

|  | (a) Observation | (b) Language Reasoning | (c) Action Reasoning |

*Figure 1.* **Language Reasoning vs. Action Reasoning.** Given an RGB image as the observation (a), the language model (b) is asked to describe the content in the image, and produces *Sentence 1* and *Sentence 2*. These sentences are flexibly ordered and jointly contribute to the global meaning required to answer the question. In contrast, the VLA model (c), when instructed to place the cup onto the plate, generates an action trajectory composed of semantically meaningful stages (*Reach→Grasp→Transport→Place*), which follow a strict order and vary in difficulty (with the more challenging stages shown in bold).

end-effector. A stage calculator computes a stage cost and per-step rewards to evaluate *how well* each stage is executed. In this way, STARE not only annotates stage-wise actions but also assesses partial successes and failures within a trajectory. We leverage STARE for offline fine-tuning via *Stage-Aware Trajectory-Wise Preference Optimization* (STA-TPO), which constructs pairwise preferences at the stage level. By incorporating stage costs, STA-TPO propagates precise gradient signals to specific action stages, enabling progressive learning and credit assignment not only between success and failure but also among varying degrees of success. For online fine-tuning, we introduce STARE to *Stage-Aware Proximal Policy Optimization* (STA-PPO), which reshapes sparse terminal rewards to dense interaction rewards. By providing this progressive feedback, STA-PPO stabilizes intra-stage updates, especially for complex manipulation tasks that require dense guidance. Conceptually, STA-TPO and STA-PPO are reminiscent of curriculum learning (Bengio et al., 2009), where training is organized along an ordered sequence of subtasks to ease optimization and improve generalization. However, unlike conventional curricula that progress strictly from easy to hard, our stage-aware design enforces semantic continuity across stages, ensuring that optimization respects causal dependencies in stages.

To sufficiently fine-tune a pre-trained VLA model with STA-TPO and STA-PPO, we integrate these two algorithms with SFT as an initialization into a serial tri-step fine-tuning pipeline, *Imitation → Preference → Interaction* (IPI). The IPI framework first finetunes a VLA model from expert demonstrations via SFT, then further optimizes it according to offline stage-aware preferences using STA-TPO, and finally refines it based on stage-aware interaction in online environments using STA-PPO.

Our contributions are summarized as: **(i)** We design STARE, a rule-based module that decomposes trajectories into se-

mantically meaningful stages, enabling fine-grained supervision beyond trajectory-level signals. **(ii)** Based on (i), we propose stage-aware fine-tuning methods: STA-TPO for offline stage-wise preference alignment and STA-PPO for online intra-stage interaction, both providing more precise credit assignment and improved sample efficiency. **(iii)** We unify supervised fine-tuning, STA-TPO, and STA-PPO into IPI, a serial tri-step pipeline for fine-tuning VLA models, and validate it on the benchmarking frameworks SimplerEnv and ManiSkill3, showing that IPI achieves state-of-the-art success rates.

## 2. Preliminary

### 2.1. Problem Formulation

We consider a language-conditioned POMDP problem defined by the tuple $\{\mathcal{S}, \mathcal{A}, \mathcal{T}, \mathcal{L}, \mathcal{R}, \gamma\}$, where $\mathcal{S}$ is the state space, $\mathcal{A}$ is the action space, $\mathcal{T} : \mathcal{S} \times \mathcal{A} \to \mathcal{S}$ is the dynamic function, $\mathcal{L}$ is the space of language instruction, $\mathcal{R} : \mathcal{S} \times \mathcal{L} \to \mathbb{R}$ is the reward function, and $\gamma$ is a scale factor with $0 < \gamma < 1$. The goal of a VLA model is to find a policy $\pi_\theta : \mathcal{S} \times \mathcal{L} \to \mathcal{A}$, which generates action trajectories maximizing the expected accumulated reward, or return for each task $l$, i.e. $\mathcal{R}(\pi, l) = \mathbb{E}_{a\sim\pi}[\sum_t \gamma^t r_t]$.

Fine-tuning a VLA model adapts a pre-trained $\pi_\theta$ to new tasks so that the resulting policy $\pi_{\theta'}$ maximizes expected return under the POMDP. This can be done through imitation for aligning with expert demonstrations, preference for refining trajectories via learned comparisons, or reinforcement optimizing long-term rewards.

### 2.2. Trajectory-Wise Preference Optimization (TPO)

Direct Preference Optimization (DPO) (Rafailov et al., 2023) is a recent fine-tuning technique for large language models that directly aligns a policy with preference data, bypassing explicit reward modeling. Extending this idea to

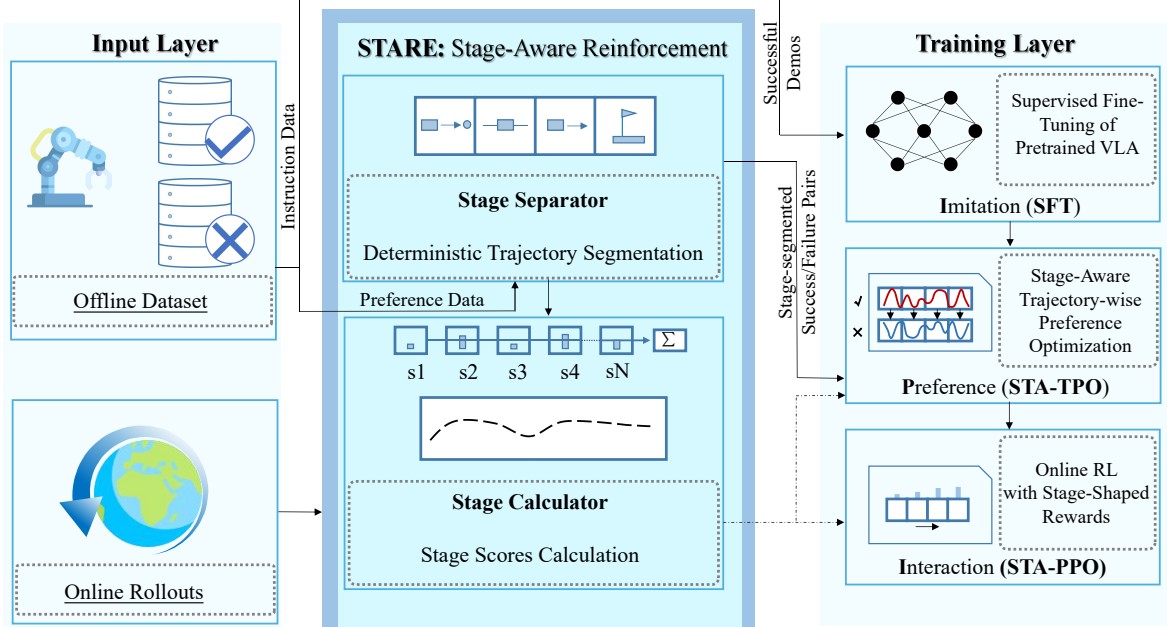

*Figure 2.* Overview of the STARE Framework and Its Integration into the IPI Training Pipeline.

fine-tuning VLA models yields TPO (Zhang et al., 2024): the outputs are action trajectories $\tau = \{(s_t, a_t)\}_{t=1}^{T}$ rather than text sequences. TPO treats each trajectory as a single sequence and learns from pairwise comparisons of successful and failed trajectories $(\tau^+, \tau^-)$ generated under the same instruction. The policy is updated to prefer $\tau^+$ over $\tau^-$ by minimizing

$$L_{\text{TPO}}(\theta) = -\mathbb{E}_{(\tau^+,\tau^-)}\Big[\log \sigma\big(\beta(q(\tau^+) - q(\tau^-))\big)\Big],$$
(1a)

$$q(\tau) = \frac{1}{T}\sum_{t=1}^{T}\Big(\log \pi_{\theta'}(a_t|s_t) - \log \pi_{\theta}(a_t|s_t)\Big).$$
(1b)

where $\sigma(\cdot)$ is the sigmoid function, $\beta$ controls the strength of preference alignment, $s_t \in \mathcal{S}$ and $a_t \in \mathcal{A}$ denote the environment state and action at timestep $t$, and $q(\cdot)$ measures the normalized log-likelihood ratio of a trajectory under policy $\pi_{\theta'}$ relative to $\pi_\theta$. $L_{\text{TPO}}$ is minimized when the model increases $q(\tau^+)$ relative to $q(\tau^-)$, i.e., when the likelihood of successful trajectories exceeds failed ones.

While TPO provides a direct mechanism to apply preference learning for VLA, it suffers from credit assignment ambiguity: preferences are assigned to full trajectories, making it difficult to determine which specific stage contributed to the preference signal (Christiano et al., 2023; Choi et al., 2024; Shen et al., 2026). Moreover, such a binary preference limits optimization to coarse distinctions between successful and failed rollouts, without capturing relative quality among

partially successful trajectories. These limitations motivate (STA-TPO), which decomposes trajectories into stages and aligns hierarchical preferences at the stage level, enabling finer-grained optimization.

### 2.3. Proximal Policy Optimization (PPO)

PPO (Schulman et al., 2017) is one of the most widely used online reinforcement learning algorithms, known for its balance of sample efficiency and training stability. PPO improves policy gradient methods by introducing a clipped surrogate objective that prevents excessively large policy updates, thereby stabilizing training. Given an old policy $\pi_\theta$, the clipped objective is

$$L_{\text{PPO}}(\theta) = \mathbb{E}_t\Big[\min\big(p_t(\theta)\,\text{GAE}(r_t),$$
$$\text{clip}\big(p_t(\theta), 1-\epsilon, 1+\epsilon\big)\,\text{GAE}(r_t)\big)\Big]$$
(2)

where $p_t(\theta) = \pi_{\theta'}(a_t|s_t)/\pi_\theta(a_t|s_t)$ is the likelihood ratio between the new and old policies, and $\epsilon$ is a clipping parameter. $\text{GAE}(\cdot)$ is generalized advantage estimator (Schulman et al., 2015) that estimate the advantage value based on rewards $r_t$.

In the context of fine-tuning VLA models, PPO is commonly used to fine-tune policies with sparse $r_t$, but such signals often limit sample efficiency and provide insufficient guidance for complex, long-horizon tasks. This motivates STA-PPO, which integrates stage-aware reward shaping to transform sparse terminal rewards into dense progressive signals for more efficient fine-tuning.

# 3. Method

## 3.1. Stage-Aware Reinforcement (STARE)

We propose STARE, a module that decomposes long-horizon action trajectories into semantically meaningful stages. STARE consists of two components: (i) a *stage separator*, which determines *when* stage transitions occur by detecting task-relevant events, and (ii) a *stage calculator*, which evaluates *how well* each stage is executed using stage-wise costs and dense intra-stage rewards.

**Stage Separator.** Stage boundaries are determined by semantically meaningful manipulation events rather than arbitrary temporal cuts. Given the whole action trajectory $\tau$, we intend to divide it into $K$ stages by defining semantic boundaries and assigning each global timestep $t$ a stage label $k \in \{1, \dots, K\}$. Following an event-driven rule, the entry condition of stage $k$ coincides with the terminal condition of stage $k - 1$, ensuring progressive continuity across stages. For instance, a pick-and-place task can be separated into four stages: *Reach → Grasp → Transport → Place*.

Stage segmentation thus reduces to detecting the onset of each stage based on geometric constraints, defined by thresholds $\delta_k$ on the translation and orientation signals of the end-effector. These thresholds set binary environment flags (e.g., grasped, on-target). For example: a *Reach → Grasp* transition occurs when the end-effector contacts the object; *Grasp → Transport* occurs when the grasped object is lifted above a small height threshold; *Transport → Place* occurs when the object is within a distance margin of the goal position; and *Place → Success* occurs when the object is released and remains stably in the goal region (see Appendix D for other segmentation examples). Thereby, we group steps with the same stage label into the $k$-th trajectory segment $\tau^{(k)} = \{(s_t, a_t) \mid g(t) = k\}_{t=1}^{T_k}$, where $s_t \in \mathcal{S}$, $a_t \in \mathcal{A}$, and $T_k$ is the number of timesteps assigned to stage $k$. Here, $g : \mathbb{N} \to \{1, \dots, K\}$ is a stage assignment function mapping each timestep $t$ to its corresponding stage index $k$. The full trajectory can then be expressed as the stage-wise decomposition $\tau \mapsto \{\tau^{(i)}\}_{i=1}^{K}$.

**Stage Calculator.** Given the stage segments produced by the stage separator, the stage calculator computes both stage-wise costs and intra-stage dense rewards by measuring the relation between the end-effector and relevant targets. The specific forms of cost and reward depend on the goal of each stage. We illustrate with *Reach* as the $k$-th stage:

(i) *Stage cost aggregation.* We define the cost function $\ell_k(\cdot)$ as the mean Euclidean distance over $T_k$ between the end-effector and the target object from start to the end of

*Reach*:

$$\ell_k(\tau^{(k)}) = \frac{1}{T_k} \sum_{t=1}^{T_k} \|x_{\text{ee}}(t) - x_{\text{obj}}(t)\|_2, \qquad (3)$$

where $x_{\text{ee}}(t) \in \mathbb{R}^3$ denotes the Cartesian position of the end-effector at time step $t$, and $x_{\text{obj}}(t) \in \mathbb{R}^3$ is the target position of the object. By definition, $\ell_k$ is a non-negative value measuring the deviation from the target: the better the $\tau^{(k)}$, the smaller the $\ell_k$. Detailed cost functions for other stage categories are provided in Appendix D.

(ii) *Intra-stage reward shaping.* To provide dense guidance, we adopt potential-based reward shaping (Kim et al., 2025a; Ng et al., 1999). For active stage $k$, we define a per-timestep potential $\Phi_{k_t}$ that captures the normalized progress of state $s_t$. Specifically, for *Reach*, we use:

$$\Phi_{k_t}(s_t) = \sigma\Big(1 - \frac{\|x_{\text{ee}(t)} - x_{\text{obj}(t)}\|}{d_k}\Big), \qquad (4)$$

where $\Phi_{k_t}(s_t) \in [0, 1]$, $d_k$ is a normalization length scale, and $\sigma(\cdot)$ is a sigmoid function. This provides smooth shaping rewards that encourage the end-effector to progressively reach the target (see detailed potential functions for other stages in Appendix D). Based on $\Phi_{k_t}$, the shaped reward $r'_t$ augments the sparse reward $r_t$ as:

$$r'_t = r_t + \gamma\, \Phi_{k_{t+1}}(s_{t+1}) - \Phi_{k_t}(s_t). \qquad (5)$$

**Remark.** STARE is model-agnostic and can be naturally extended to learned event predictors or neural stage classifiers, enabling end-to-end stage discovery.

## 3.2. From STARE to STA-TPO

Standard TPO (Zhang et al., 2024) aggregates preferences only at the level of entire trajectories, STA-TPO leverages STARE to segment trajectories into progressive stages and perform stage-wise preference alignment. A detailed algorithm is provided in A.1. A pair comparison of stage samples $(\tau^{(k)+}, \tau^{(k)-})$ exists only when the previous stage $\tau^{(k-1)}$ has been successfully completed, ensuring progressive consistency across stages. In addition, the stage cost $\ell_k(\tau)$ is incorporated as a penalty term in (1b), transforming $q$ into $\hat{q}$:

$$\hat{q}(\tau^{(k)}) = q(\tau^{(k)}) - \lambda \ell_k(\tau^{(k)}), \qquad (6)$$

where $\lambda$ is the penalty weight. The original objective $\mathcal{L}_{\text{TPO}}$ in (1a) thereby extends to $\mathcal{L}_{\text{STA-TPO}}$. Compared to (1), which optimizes the model only with binary trajectory-level preferences (success vs. failure), $\hat{q}$ introduces a hierarchical signal to $\mathcal{L}_{\text{STA-TPO}}$. Even among successful stages $\tau^{(k)+}$ across different trajectories, those with lower penalties $\ell_k(\tau^{(k)})$ yield higher $\hat{q}$, while less optimal stages receive lower $\hat{q}$. This design enables credit assignment not only between success and failure but also among varying degrees of success, thereby providing finer-grained supervision for learning optimal behaviors.

*Table 1.* **Evaluation on SimplerEnv and ManiSkill3.** Left: SimplerEnv-Bridge tasks (Success %). Right: ManiSkill3-Franka tasks (Success %). OpenVLA-7B and Pi0.5 based methods use 100 trajectories for SFT and 50 preference pairs for TPO and STA-TPO. The '(+X)' indicates the absolute improvement in success rate (%) over a relevant baseline.

| Methods | SimplerEnv-Bridge | | | | | ManiSkill3-Franka | | | | |
|---|---|---|---|---|---|---|---|---|---|---|
| | Put Spoon on Towel | Put Carrot on Plate | Stack Green on Yellow | Put Eggplant in Basket | Avg. Success | Stack Cube | Push Cube | Pull Cube | LiftPeg Upright | Avg. Success |
| *OpenVLA-7B (Kim et al., 2024) Based* | | | | | | | | | | |
| SFT | 43.7 | 52.7 | 21.3 | 49.0 | 41.7 | 12.0 | 11.7 | 31.0 | 5.3 | 15.0 |
| GRAPE (Zhang et al., 2024) | 44.3 | 55.0 | 22.7 | 53.7 | 43.9 | 15.7 | 13.3 | 35.3 | 7.7 | 18.0 |
| SFT → STA-TPO (STARE) | 51.0 | 57.3 | 43.7 | 54.3 | 51.6 (+17.7) | 19.3 | 16.0 | 35.7 | 12.3 | 20.8 (+2.8) |
| RL4VLA (Liu et al., 2025) | 93.0 | 91.3 | 92.0 | 93.7 | 92.5 | 64.0 | 95.7 | 90.3 | 32.0 | 70.5 |
| SFT → STA-PPO (STARE) | 94.3 | 95.3 | 93.7 | 95.0 | 94.6 (+2.1) | 92.7 | 96.0 | 95.3 | 89.7 | 93.4 (+22.9) |
| **IPI (STARE)** | **98.0** | **98.5** | **98.0** | **97.5** | **98.0** | **94.3** | **97.3** | **98.5** | **95.5** | **96.4** |
| *Pi0.5 (Black et al., 2025) Based* | | | | | | | | | | |
| SFT | 49.3 | 64.7 | 44.7 | 69.7 | 57.1 | 26.3 | 18.3 | 43.0 | 10.7 | 24.6 |
| GRAPE (Zhang et al., 2024) | 48.0 | 59.3 | 48.3 | 58.7 | 53.6 | 22.7 | 16.3 | 45.0 | 6.3 | 22.6 |
| SFT → STA-TPO (STARE) | 54.0 | 65.3 | 54.0 | 68.7 | 60.5 (+6.9) | 28.0 | 22.3 | 44.7 | 15.7 | 27.7 (+5.1) |
| $\pi_{RL}$ (Chen et al., 2025a) | 82.7 | 97.3 | 83.3 | 55.0 | 79.6 | 72.3 | 96.7 | 93.3 | 58.0 | 80.1 |
| SFT → STA-PPO (STARE) | 90.7 | 97.7 | 85.7 | 63.7 | 84.5 (+4.9) | 80.7 | 98.0 | 92.7 | 75.3 | 86.7 (+6.6) |
| **IPI (STARE)** | **95.7** | **98.7** | **93.0** | **78.7** | **91.5** | **84.3** | **99.3** | **95.0** | **80.7** | **89.9** |

## 3.3. From STARE to STA-PPO

For online RL fine-tuning, we integrate STARE directly into rollouts. The stage separator determines the stage transition online. At each time step within the stage, the stage calculator produces shaped reward $r'_t$, turning $L_{PPO}$ in (2) into $L_{STA\text{-}PPO}$. Finally, policy parameters $\theta$ are updated by minimizing $L_{STA\text{-}PPO}$. By replacing $r_t$ with $r'_t$, STA-PPO provides denser, stage-aligned feedback that accelerates policy learning in long-horizon, sparse-reward tasks.A detailed algorithm is provided in Appendix A.2.

## 3.4. STA-TPO And STA-PPO for Serial Fine-tuning

Existing works often apply offline preference-based optimization (Zhang et al., 2024) and online RL fine-tuning (Liu et al., 2025; Li et al., 2025b) separately. Besides, while we are now able to jointly address offline preference alignment and online reinforcement learning by STA-TPO and STA-PPO, a complete fine-tuning framework for VLA models must also incorporate imitation learning to initialize a strong policy prior.

Thereby, we propose Imitation → Preference → Interaction (IPI), a three-step fine-tuning pipeline. We first warm up the policy from demonstrations by SFT. Then we apply STA-TPO to offline refine the policy. Finally, we apply STA-PPO to further enhance robustness through online exploration. Thereby, IPI integrates supervised, preference-based, and exploration signals into a coherent progression, yielding more sample-efficient and more robust fine-tuning of VLA models.

## 4. Experiments

### 4.1. Simulation Experiments

**Benchmarks.** We evaluate our approach on two families of robotic manipulation environments, as illustrated in Appendix C.1. The first is **SimplerEnv** (Li et al., 2024) with the WidowX arm, where we focus on the four canonical single-object tasks in the **SimplerEnv-Bridge** split. The second is **ManiSkill3** (Tao et al., 2025) with the Franka robot (Haddadin, 2024), including *StackCube* and three contact-rich tasks (*PushCube*, *PullCube*, and *LiftPegUpright*) to validate generality beyond pick-and-place and assess performance under challenging non-trivial manipulation.

**Baselines.** We compare against the strong offline preference fine-tuning method GRAPE (Zhang et al., 2024), and the RL fine-tuning baselines RL4VLA (Liu et al., 2025) and $\pi_{RL}$ (Chen et al., 2025a). For fairness, all methods fine-tune the OpenVLA-7B (Kim et al., 2024) and pi0.5_base (Black et al., 2025) backbone, and we additionally evaluate our proposed STA-TPO, STA-PPO, and the full **IPI**. We report average success over 300 evaluation episodes per method. Unless otherwise stated, hyperparameters are shared across methods when applicable (detailed in Appendix B).

**Results.** We begin by presenting overall comparisons on two families of manipulation benchmarks. Results on SimplerEnv-Bridge and ManiSkill3-Franka are shown in Table 1. Across all benchmarks, existing VLA baselines exhibit limited performance (e.g., average success rates < 60%). Recent RL fine-tuning approaches, such as RL4VLA (Liu et al., 2025) achieve strong results (92.5% on SimplerEnv-WidowX, 70.5% on ManiSkill3). Our pro-

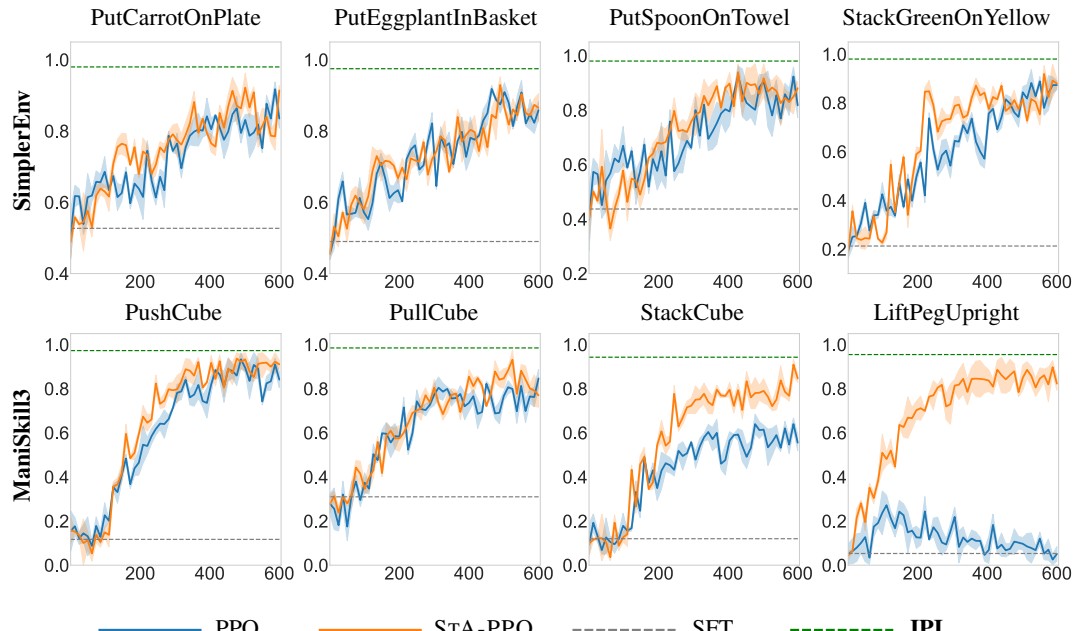

*Figure 3.* **Learning curves on SimplerEnv and ManiSkill3.** We report the mean success rate across **N=3** random seeds. Shaded regions indicate one standard deviation across seeds. The y-axis denotes the success rate, and the x-axis shows the interaction environment steps (in thousands).

posed **IPI** further improves to 98.0% and 96.4%, outperforming prior state-of-the-art methods by +5.4 and +25.9 points, nearly solving these benchmark tasks. Flow-based VLA models such as $\pi_{0.5}$ also benefit substantially from our stage-aware reinforcement design: integrating STARE consistently boosts $\pi_{0.5}$'s performance across all tasks, outperforming its original flow-matching baseline.

While PPO improves over SFT, it often stagnates on tasks requiring high precision. In contrast, STA-PPO consistently accelerates convergence and achieves higher asymptotic performance by leveraging stage-aware signals. Figure 3 presents results across eight representative tasks from SimplerEnv and ManiSkill3. The most challenging tasks—*LiftPegUpright* and *StackGreenOnYellow*—exhibit the largest performance gaps, underscoring the importance of incorporating stage-aware signals in precision-critical manipulation. By comparison, for pick-and-place tasks (e.g., *PutCarrotOnPlate*, *PutEggplantInBasket*) or simple push-and-pull tasks, PPO and STA-PPO achieve similar final success rates, with STA-PPO mainly contributing faster convergence and reduced variance. Overall, these results suggest that stage-aware guidance is particularly crucial when strict alignment accuracy or multi-stage coordination is required.

**OOD Generalization** We also evaluate how STARE reacts to distribution shifts, we follow the generalization test protocol proposed by (Liu et al., 2025) and reproduce their out-of-distribution (OOD) settings on SimplerEnv as a light sanity check. This setup introduces controlled shifts along visual, semantic, and execution dimensions. This serves as a preliminary verification that the proposed method does not overfit to the training environments. Full experimental details and results are provided in Appendix E.

### 4.2. Ablations study

After the benchmark-level comparison in Table 1, we note that while the overall improvements of STA-PPO and STA-TPO over prior baselines are consistent, the performance gap is most pronounced on two tasks: (1) Cube stacking tasks from both the environments, which requires precise alignment in placing stage, and (2) *LiftPegUpright* from ManiSkill3, which demands accurate orientation control after lifting. To better understand where these gains originate, we decompose trajectories into semantic stages and evaluate **conditional stage success rate (CSSR)** ($P(\text{stage}_k \mid \text{stage}_{k-1})$), which measures how reliably a policy completes a stage given that all previous stages have been successful.

Figure 4 shows that STA-TPO provides clear advantages over TPO, with the largest improvements appearing in the **grasp**, and **place** and **upright** stages. These stages are particularly decisive for the final outcome, explaining why the overall success rate improvements are disproportionately large for these two tasks.

To dissect the contributions of STARE, we conduct a stage toggle ablation where the STARE signal is selectively removed at different phases of the manipulation in STA-PPO. As shown in Figure 5, disabling STARE at early stages (e.g.,

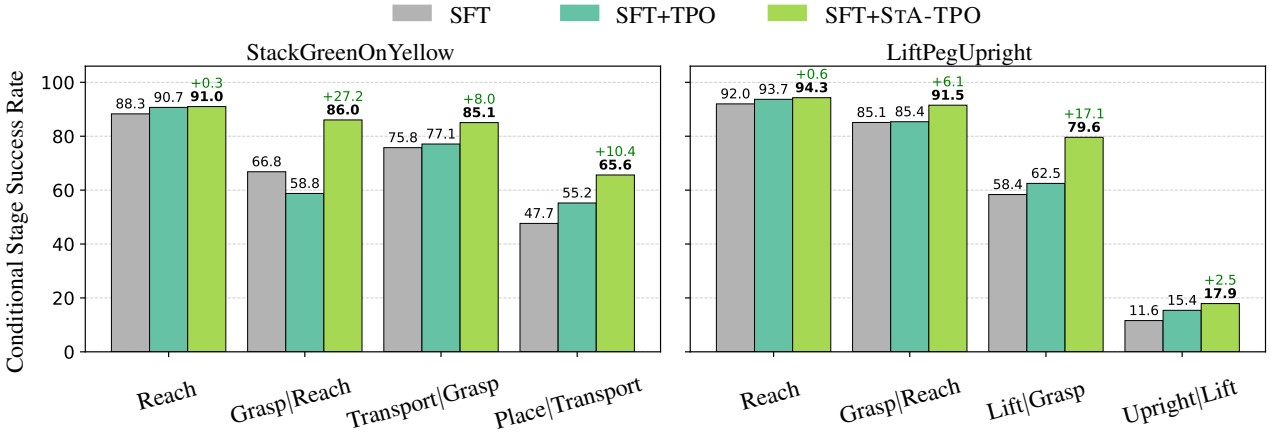

Figure 4. **Offline Stage-wise ablation on two tasks.** We report CSSR (%) for *StackGreenonYellow* (SimplerEnv) and *LiftPegUpright* (ManiSkill3). Compared with TPO, STA-TPO achieves significant gains, particularly in the **grasp** and **place/upright** stages.

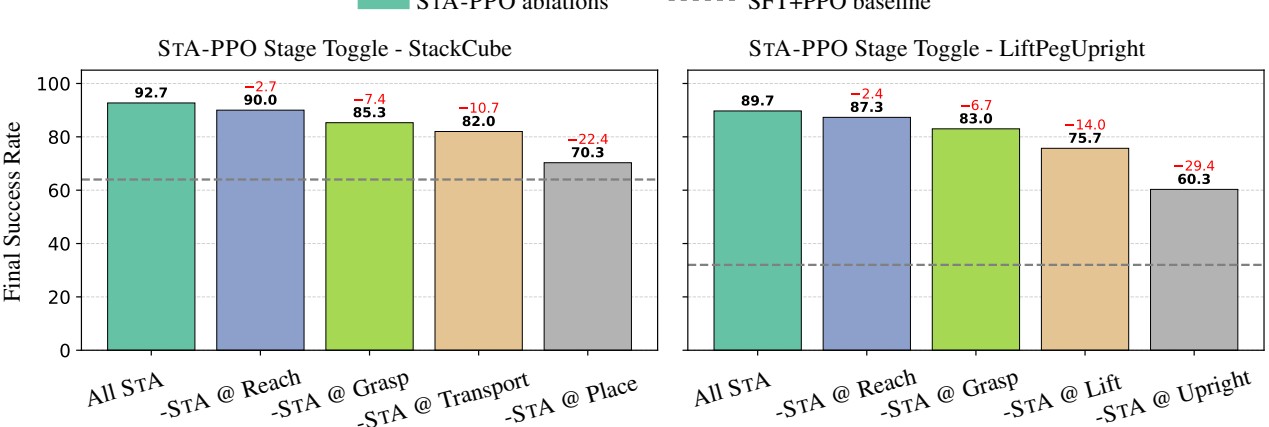

Figure 5. **Stage toggle ablation of STA-PPO.** We evaluate the effect of selectively disabling stage-aware reinforcement signals on two representative tasks: *StackCube* (ManiSkill3) and *LiftPegUpright* (ManiSkill3). The **All STA (STARE)** setting achieves the best performance, while disabling critical stages (**Place** in stacking, **Upright** in peg lifting) causes the largest performance drops.

reach or grasp) only leads to moderate drops, since later corrective actions can partially recover performance. In contrast, removing STARE at the final precision-critical phases (e.g., **Place** in stacking and **Upright** in peg lifting) causes the largest degradation, reducing success rates by more than 20%. This analysis highlights that STARE guidance is especially valuable at stages where geometric accuracy and stability directly determine task completion.

### 4.3. Real-World Experiments

**Experimental Setup.** As shown in Figure 6, we conduct real-world experiments using a UR3e robot arm equipped with a Robotiq gripper . For visual input, we deploy two RealSense D435i cameras in eye-on-hand and eye-on-base configurations to capture RGB and RGB-D observations. We fine-tune OpenVLA-7B and Pi0.5_base on a self-collected dataset of real-robot demonstrations, consisting of RGB observations, robot states, and actions, all collected via 6-DoF SpaceMouse-based teleoperation.

We use a open-vocabulary segmentation module based on Grounded SAM (Ren et al., 2024) to obtain object masks from language queries. Object masks are lifted to 3D using aligned RGB-D observations, and the resulting centroids are used as object states for computing stage-aware rewards. For the *place* stage, we localize the target basket using STag (Benligiray et al., 2019) to define the placement goal. With this set up, we evaluate our methods on two manipulation tasks: Red Apple (*"Put the red apple in the basket."*) and Stack Cups (*"Stack the green cup on the yellow cup."*). Each method is evaluated on 30 independent trials per task, with randomized initial object poses and robot starting configurations. Detailed in Appendix C.1.3.

**Results.** As shown in Table 2, our method STA-TPO outperforms DPO and GRAPE on all real-world tasks, achieving the highest success rates on Red Apple and Stack Cups. More importantly, stage-wise analysis reveals consistent improvements in CSSR at all manipulation stages. This

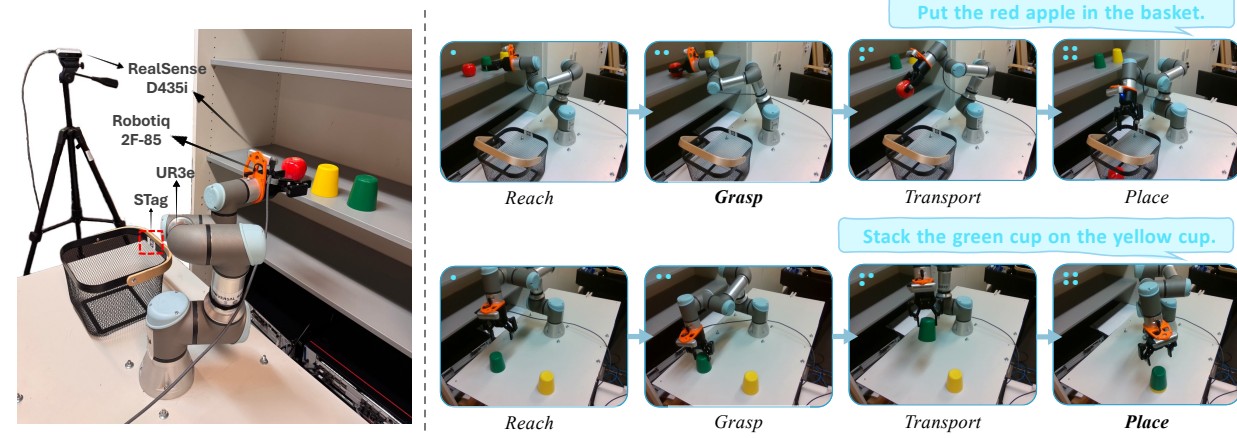

*Figure 6.* **Real-robot setup and stage-wise execution tasks.**

*Table 2.* **Real-world manipulation evaluation.** We report CSSR (%) for the four manipulation stages, and the overall task success rate (SR, %). Specifically, CSSR corresponds to $P(\text{Reach})$, $P(\text{Grasp} \mid \text{Reach})$, $P(\text{Transport} \mid \text{Grasp})$, and $P(\text{Place} \mid \text{Transport})$, respectively. The '(+X)' indicates the absolute improvement in success rate (%) over a relevant baseline.

| Method | Red Apple (4 Stages) | | | | | Stack Cups (4 Stages) | | | | |
|---|---|---|---|---|---|---|---|---|---|---|
| | 1 | 2 | 3 | 4 | Overall | 1 | 2 | 3 | 4 | Overall |
| *OpenVLA-7B (Kim et al., 2024) Based* | | | | | | | | | | |
| SFT | 73.3 | 36.4 | 62.5 | 80.0 | 13.3 | 80.0 | 41.7 | 60.0 | 33.3 | 6.7 |
| SFT → DPO (Zhang et al., 2024) | 86.7 | 46.2 | 58.3 | 71.4 | 16.7 | 76.7 | 47.8 | 60.0 | 45.5 | 10.0 |
| GRAPE (Zhang et al., 2024) | 90.0 | 55.6 | 80.0 | 83.3 | 33.3 | 96.7 | 55.2 | 76.9 | 40.0 | 13.3 |
| SFT → STA-TPO (**STARE**) | **96.7** | **58.6** | **88.2** | **86.7** | **43.3** (+10.0) | 93.3 | **75.0** | **85.7** | **55.6** | **36.7** (+23.4) |
| *Pi0.5 (Black et al., 2025) Based* | | | | | | | | | | |
| SFT | 86.7 | 46.2 | 80.0 | 83.3 | 26.7 | 83.3 | 52.2 | 92.0 | 33.3 | 13.3 |
| SFT → DPO (Zhang et al., 2024) | 93.3 | 57.1 | 87.5 | 85.7 | 40.0 | 86.7 | 53.8 | 78.6 | 45.5 | 16.7 |
| GRAPE (Zhang et al., 2024) | 90.0 | 55.6 | 86.7 | 84.6 | 36.7 | 93.3 | 57.1 | **93.8** | 46.7 | 23.3 |
| SFT → STA-TPO (**STARE**) | **96.7** | **79.3** | **91.3** | **90.5** | **66.7** (+30.0) | **100** | **66.7** | 90.0 | **50.0** | **30.0** (+6.7) |

indicates that STARE provides finer-grained and more effective credit assignment than trajectory-level baselines.

We further observe task-specific failure patterns that align well with our simulation findings. For *Red Apple* task, the grasp stage exhibits the lowest CSSR, suggesting grasping as the primary bottleneck. In *Stack Cups* task, the place stage is the most challenging, due to the increased precision required for stacking. The consistency between simulation and real-world stage-wise behaviors validates the generality and interpretability of our STARE framework.

## 5. Related Works

Our work is closely related to Long-horizon Robotic Manipulation Tasks in RL, RL Fine-tuning for LLMs, and RL Fine-tuning for VLAs. A comprehensive discussion is provided in Appendix F.

## 6. Conclusion

We presented *Stage-Aware Reinforcement* (STARE), a plug-in module that decomposes trajectories into semantically

meaningful stages and provides stage-level reinforcement signals. Building on this, we introduced Stage-Aware TPO (STA-TPO) and PPO (STA-PPO) for offline stage-wise preference alignment and online intra-stage interaction, and integrated them with supervised fine-tuning into the *Imitation → Preference → Interaction* (IPI) pipeline. Experiments on SimplerEnv and ManiSkill3 demonstrate that IPI achieves state-of-the-art success rates.

Extensive experiments on both OpenVLA and Pi0.5 backbones, across SimplerEnv, ManiSkill3, and real-world robotic tasks, show that STARE consistently yields substantial gains over trajectory-level baselines. These results suggest that stage-aware credit assignment provides a more practical optimization signal for VLA fine-tuning, as it improves training stability, accelerates convergence, and yields consistent gains under distribution shifts and real-world deployment. We hope this stage-centric view will encourage future work to move beyond trajectory-level supervision toward more structured and interpretable learning signals for VLA models.

## Impact Statement

This paper presents work whose goal is to advance the field of Machine Learning, specifically in robotics and vision-language-action models for embodied AI. Our method improves the sample efficiency and performance of robotic manipulation through stage-aware reinforcement learning. The proposed techniques enable robots to learn robotic manipulation tasks more effectively, which could contribute to safer and more capable autonomous systems in real-world applications. While our experiments are conducted in simulation and controlled real-world settings, future deployment of such systems should consider safety, reliability, and ethical implications of autonomous robotic systems. There are many potential societal consequences of our work, none which we feel must be specifically highlighted here beyond standard considerations for robotics research.

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

## A. Detailed Algorithms

---

**Algorithm 1** STA-TPO (offline)

---

**Require:** Preference pairs $\{(\tau^+, \tau^-)\}$ under same instruction; reference policy $\pi_\theta$; STARE; stage penalty weight $\lambda$; temperature $\beta$; learning rate $\eta$.

**Ensure:** Updated policy $\pi_{\theta'}$.

1: **while** not converged **do**
2:    Sample minibatch $\mathcal{B} = \{(\tau^+, \tau^-)\}$.
3:    **for all** $\tau \in \{\tau^+, \tau^-\}$ in $\mathcal{B}$ **do**
4:       **Stage segmentation & costs:** $\{\tau^{(k)}, \ell_k(\tau^{(k)})\}_{k=1}^K \leftarrow$ STARE$(\tau)$.
5:       **Stage scores:** For each $k$, first compute

$$q(\tau^{(k)}) \leftarrow \frac{1}{T_k} \sum_{t \in T_k} \Big( \log \pi_{\theta'}(a_t|s_t) - \log \pi_\theta(a_t|s_t) \Big). \qquad \text{(cf. equation 1b)}$$

6:       Then add stage costs: $\hat{q}(\tau^{(k)}) \leftarrow q(\tau^{(k)}) - \lambda\, \ell_k(\tau^{(k)})$.
7:    **end for**
8:    **Preference loss:** Compute

$$L_{\text{STA-TPO}} \leftarrow -\frac{1}{|\mathcal{B}|} \sum_{(\tau^+, \tau^-) \in \mathcal{B}} \frac{1}{K} \sum_{k=1}^K \log \sigma\Big( \beta\big(\hat{q}(\tau^{(k)+}) - \hat{q}(\tau^{(k)-})\big) \Big). \qquad \text{(cf. equation 1a)}$$

9:    **Policy update:** $\theta' \leftarrow \theta' - \eta\, \nabla_{\theta'} L_{\text{STA-TPO}}$.
10: **end while**=0

---

## B. Training Settings and Evaluation Protocols

### B.1. Supervised Fine-Tuning (SFT)

We initialize SFT models from two pretrained VLA backbones: OpenVLA-7B and the pi0.5_base. For each backbone, we optimize the action prediction objective using the AdamW optimizer with a learning rate of $1 \times 10^{-5}$ and a constant schedule. SFT is trained for 100K steps per backbone on each benchmark, and the same training budget is used for all methods on the same backbone to ensure fair comparison. All image observations are resized to $224 \times 224$ before being fed into the VLA backbone.

### B.2. Trajectory Preference Optimization: TPO and STA-TPO

For offline preference optimization, we start from SFT checkpoints of both OpenVLA-7B and pi0.5_base backbone, and construct datasets of paired successful ("chosen") and failed ("rejected") trajectories. Each trajectory consists of image observations, language instructions, and continuous robot actions, which are discretized into tokens using an action tokenizer built on top of the base vocabulary.

We use a DPO-style preference objective that increases the log-likelihood ratio between chosen and rejected trajectories relative to a frozen SFT reference model. This provides a stable and scalable way to perform trajectory-level preference alignment. Training uses AdamW with a learning rate of $2 \times 10^{-5}$, gradient accumulation, and 50K optimization steps per backbone to ensure fair comparison across baselines.

STA-TPO uses the same data pipeline, objective, and optimizer, but modifies the DPO logit difference by adding stage-aware margins computed from our stage calculator. This biases the preference updates toward stages where the chosen and rejected trajectories differ the most, while keeping easy stages largely unchanged, yielding a stage-aware variant that is directly comparable to standard TPO.

---

**Algorithm 2** STA-PPO (online)

---

**Require:** Simulation Env; Behavior policy $\pi_\theta$; STARE; horizon $T$; PPO epochs $E$; discount $\gamma$; GAE parameter; clip $\epsilon$; step size $\eta$.

**Ensure:** Updated policy $\pi_{\theta'}$.

1: **while** not converged **do**

2:    **Rollout:** collect $\{(s_t, a_t, r_t, \log \pi_\theta(a_t|s_t))\}_{t=0}^{T-1}$ in Env.

3:    **Online stage labels & potentials:**

4:    **for** $t = 0$ to $T - 1$ **do**

5:       Detect current stage $k = g(t)$ via stage separator in STARE (event rules).

6:       Compute potential $\Phi_{k_t}(s_t)$ by the stage calculator in STARE.

7:    **end for**

8:    **Shaped rewards:**

9:    **for** $t = 0$ to $T - 1$ **do**

10:      $r_t' \leftarrow r_t + \gamma \, \Phi_{k_{t+1}}(s_{t+1}) - \Phi_{k_t}(s_t)$ {Potential-based shaping}

11:    **end for**

12:    **for** $e = 1$ to $E$ **do**

       {PPO updates}

13:       Compute ratio $p_t(\theta') = \exp(\log \pi_{\theta'}(a_t|s_t) - \log \pi_\theta(a_t|s_t))$.

14:       **Interaction loss:**

$$L_{\text{STA-PPO}} \leftarrow \mathbb{E}_t \big[ \min \big( p_t(\theta') \, \text{GAE}(r_t'), \; \text{clip}(p_t(\theta'), 1 - \epsilon, 1 + \epsilon) \, \text{GAE}(r_t') \big) \big] . \qquad \text{(cf. equation 2)}$$

15:       **Update:** $\theta' \leftarrow \theta' + \eta \, \nabla_{\theta'} L_{\text{STA-PPO}}$.

16:    **end for**

17: **end while**=0

---

## B.3. Reinforcement Learning: PPO and STA-PPO

We adopt an on-policy PPO framework. For SimplerEnv, we run 100 parallel environments with an episode horizon of 60 steps, yielding 6000 transitions per PPO update. Rollouts are stored in a separated replay buffer, and advantages are computed using generalized advantage estimation with a discount factor $\gamma = 0.99$ and GAE parameter $\lambda = 0.95$.

Optimization follows the standard clipped PPO objective. We use the AdamW optimizer with a policy learning rate of $1 \times 10^{-4}$, a value learning rate of $3 \times 10^{-3}$, and gradient accumulation over 20 steps to stabilize large-model training. Each update performs one PPO epoch over four minibatches, and the entropy coefficient is set to 0.0. Unless otherwise stated, ManiSkill3 uses the same optimization settings and training budget. All runs are trained for 600K environment steps to ensure fair comparison across baselines.

STA-PPO follows the same training pipeline and computational budget but augments the reward with dense stage-aware signals generated by the stage calculator. These structured shaping terms provide intermediate guidance that complements sparse task rewards, and the system additionally logs per-stage performance statistics during rollouts. Aside from this reward augmentation, STA-PPO is identical to PPO, enabling controlled comparisons of stage-aware credit assignment.

## B.4. Evaluation Protocols

We evaluate all methods on both SimplerEnv-WidowX and ManiSkill3 benchmarks under a unified protocol. Each policy is tested over 300 evaluation episodes with deterministic decoding. A rollout is considered successful if the environment-defined task condition is satisfied within the evaluation horizon (60 steps for SimplerEnv and 30 steps for ManiSkill3). Results are averaged over three random seeds. In addition to final success rates, we compute conditional stage success to diagnose at which manipulation stages policies succeed or fail.

## B.5. Checkpoint Selection

We adopt a fixed-duration training schedule for all methods and use a consistent checkpoint-selection protocol across backbones and benchmarks.

*Figure 7.* Two simulated benchmarks. We show experiment setups and example tasks involved.

For TPO and STA-TPO, models are trained for 50K optimization steps, and evaluated every 1,000 training steps on both SimplerEnv and ManiSkill3. The best-performing checkpoint under this evaluation protocol is reported.

For PPO and STA-PPO, models are trained for 600K steps. We run periodic evaluations every 6,000 environment steps on SimplerEnv and every 3,000 environment steps on ManiSkill3, selecting the checkpoint with the highest success rate.

For all ablation studies, we fix the total training duration and report the final checkpoint to ensure strict comparability across variants.

## C. Experimental Setup and Implementation Details

### C.1. Environments and Task Definitions

To evaluate robotic policies under diverse manipulation scenarios, we conduct experiments in two simulation environments: **SimplerEnv-Bridge** and **ManiSkill3-Franka**, Figure 7 provides an overview of both benchmarks, including the robot platforms and representative tasks.

#### C.1.1. SIMPLERENV-BRIDGE

We adopt the benchmark provided by SimplerEnv (Li et al., 2024). Within this environment, we test the following tasks:

- **Put the spoon on the towel** — The spoon is initialized at one corner of a $15 \times 15\,\text{cm}^2$ square region on the tabletop; a towel is placed at the opposite corner. The spoon's orientation alternates between horizontal and vertical across trials, requiring the gripper to adapt its pose accordingly.

- **Put the carrot on the plate** — This task mirrors the previous one, with a carrot replacing the spoon and a plate replacing the towel.

- **Stack the green block on the yellow block** — A green block and a yellow block (both 3 cm cubes) are placed at different corners of a tabletop square. We evaluate two square sizes (side length 10 cm and 20 cm). This task tests precise grasping, lifting, and alignment to form a stable stack.

- **Put the eggplant into the yellow basket** — An eggplant is randomly placed in the right basin of a simulated sink, and a yellow basket is placed in the left basin. The eggplant's position and orientation vary across trials (while avoiding basin edges to ensure graspability). This task challenges the policy's ability to deal with irregular shapes and variable object poses.

#### C.1.2. MANISKILL3-FRANKA

To further stress-test manipulation under contact-rich and long-horizon tasks, we use four tasks from ManiSkill3 (Tao et al., 2025) with a simulated Franka robot:

- **Stack Cube** — The robot must stack one cube on top of another, requiring accurate grasping, lifting, alignment, and stable placement.

- **Push Cube** — A cube is placed on the table, and the robot must push it toward a designated target region. This task emphasizes smooth trajectory execution and contact-rich control.

- **Pull Cube** — The robot grasps the cube on one side and pulls it (dragging) into the target region, demanding stable grasp maintenance under sliding contact.

- **Lift Peg Upright** — A peg is initialized lying flat on the table; the robot must grasp it, lift it, and reorient it so that it stands vertically upright. This task is challenging due to orientation constraints and the need for precise control to maintain balance.

### C.1.3. REAL ROBOT-UR3E

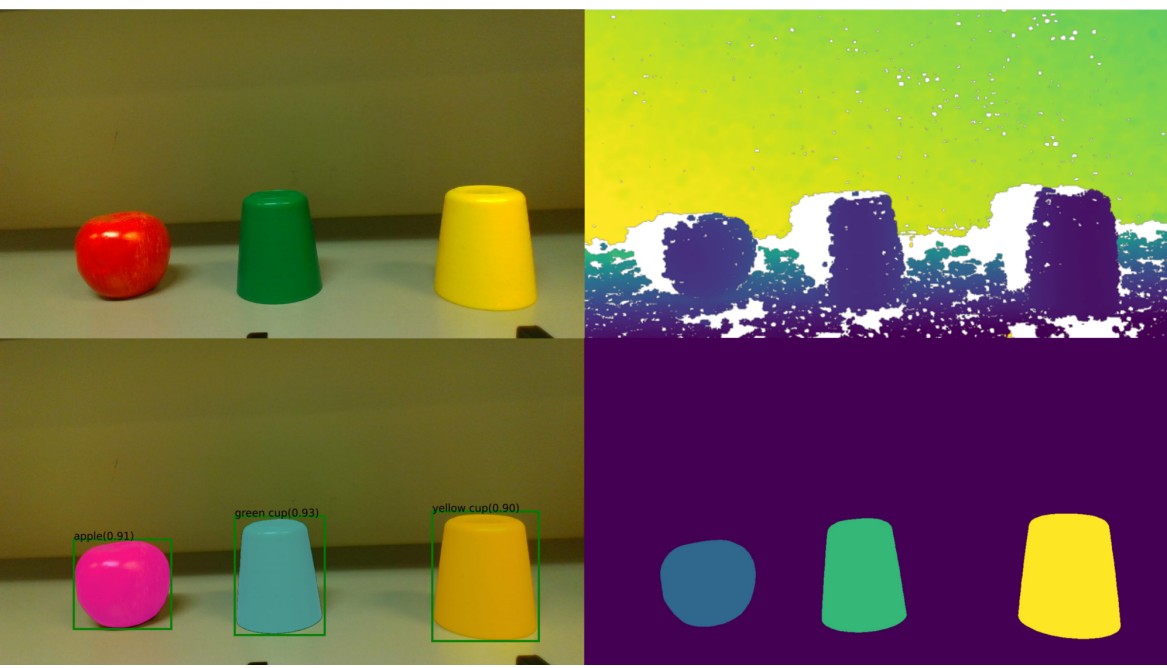

*Figure 8.* **Open-vocabulary 3D object grounding.** From left to right: **(a)** RGB observation; **(b)** RGB-D observation; **(c)** object detection and mask-based localization with class confidence; **(d)** instance-level semantic masks lifted to 3D and converted into object states.

As shown in Fig. 8, we employ Grounded SAM to detect task-relevant objects from RGB observations. The predicted masks are lifted to 3D using aligned RGB-D data, and the centroid of each instance is used as a physically grounded object state. These 3D object states enable automatic computation of stage-aware rewards based on relative geometry, without requiring manual annotations or task-specific heuristics.

For the *place* stage in Red Cub task, the target location is obtained using a fiducial marker system called STag. As illustrated in Fig. 9, we attach an STag marker to the basket to provide a stable 3D target reference. The detected marker is used to estimate the basket position, which serves as the goal state for computing object-to-target distances during placement. This design ensures reliable grounding of target semantics in real-world settings while remaining fully automatic.

**Real-World Task Specifications.**

- **Red Apple.** The robot is instructed with "Put the red apple in the basket." At the beginning of each trial, the apple is randomly placed on the table within the robot workspace, while the basket is fixed and localized using an STag marker. The task is successful if the apple is fully inside the basket and remains stable for at least 2 seconds.

- **Stack Cups.** The robot is instructed with "Stack the green cup on the yellow cup." At the start of each trial, both cups are randomly placed on the table with random planar offsets and orientations. The task is successful if the green cup is placed on top of the yellow cup and remains stable for at least 2 seconds without falling.

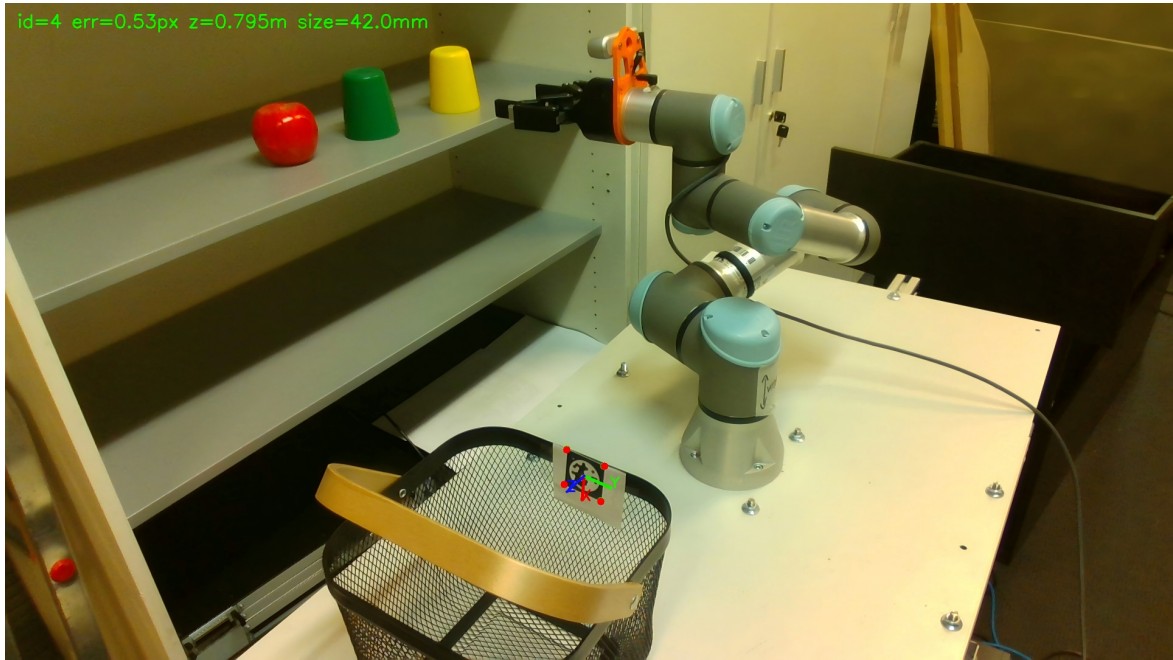

*Figure 9.* STag-based target localization.

For each task, all methods are evaluated on **30 independent trials** with randomized initial object poses and robot starting configurations. Each trial is executed for a fixed horizon and is terminated early upon success or failure.

### C.2. Evaluation Protocol

For all experiments, we follow the evaluation procedure below:

- Each task is repeated across the full set of defined trials. For SimplerEnv we run 300 trials per task; for ManiSkill3 we follow the standard configurations defined by the benchmark.

- Success is determined based on task-specific completion criteria (e.g., correct placement, stable stack, upright peg, etc.).

- For tasks (or sequences) that involve multiple steps, we additionally record the trajectory length and measure efficiency, robustness, and consistency across trials.

- Final results (success rates, trajectory lengths) are reported as mean ± standard deviation over all trials, ensuring statistical reliability.

### C.3. Implementation Details

- Both SimplerEnv and ManiSkill3 are used with their default simulation configurations. We do not modify the underlying environment logic — only vary object initial positions and orientations as described.

- For each trial, object poses (positions, orientations when applicable) are randomized within the constraints defined for each task to ensure diversity across trials.

- Policies operate in an end-to-end fashion (observations → actions), without manual heuristics or task-specific engineering.

- All reported metrics are averaged over the full set of trials. When appropriate, we also report standard deviations to reflect variability.

### C.4. Data Collection

Our data collection pipeline is designed to support supervised fine-tuning, preference-based optimization, and real-world reinforcement for long-horizon manipulation. Datasets are collected in both simulation and the real world using a unified task specification.

**Demonstration Trajectories.** For each task, we collect expert demonstration trajectories in both simulation and the real world. In simulation (ManiSkill3 and SimplerEnv), we generate **100 trajectories per task** using the MPLib motion planner or the Octo policy. In the real world, we collect **200 trajectories per task** on a UR3e robot via 6-DoF SpaceMouse teleoperation. All trajectories contain synchronized RGB-D observations, robot joint states, and executed actions. We apply an action filtering step that removes idle actions where the end-effector pose changes by less than a small threshold, which mitigates the tendency of SFT policies to stall during execution.

**Preference Trajectory Pairs.** For methods requiring preference supervision (e.g., DPO, GRAPE, STA-TPO), we generate **50 trajectory preference pairs per task**. In SimplerEnv and ManiSkill3, each pair is formed by sampling two trajectories from the demonstration pool and assigning preference labels based on task completion and cumulative reward. For real robotic experiments, we collect **100 trajectory preference pairs per task** on the UR3e robot also using 6-DoF SpaceMouse teleoperation. Our dataset focuses on two manipulation tasks, **Red Apple** and **Stack Cups**, each with four stages. Stage boundaries and stage-wise rewards are automatically inferred using the physically grounded geometry-based signals.

## D. Stage-wise Cost and Potential Definitions

This appendix provides the complete definitions of the stage-wise costs $\ell_k(\tau^{(k)})$ and progress potentials $\Phi_k(s)$ used in STARE. Each stage corresponds to a geometric manipulation primitive. Stage costs—used in STA-TPO—evaluate how well a trajectory segment achieves the geometric goal of a stage, while stage potentials—used in STA-PPO—provide dense, per-step shaping signals that reflect normalized progress within a stage. All potentials follow a normalized sigmoid form,

$$\sigma(z) = \frac{1}{1 + e^{-z}},$$

and lie in $[0, 1]$. For stages involving orientation, we additionally use a normalized rotational alignment measure $g(R_1, R_2) \in [0, 1]$, where $g(R_1, R_2)$ denotes a normalized orientation alignment measure.

**Stage Decomposition Across Tasks.** Different manipulation tasks instantiate different subsets of these generic manipulation primitives. We use a minimal, rule-based decomposition based on geometric event boundaries that can be reliably detected from state observations:

- **Pick-Place:** Reach → Grasp → Transport → Place.

- **Push:** Reach → Push → Goal.

- **Pull:** Reach → Pull → Goal.

- **Lift Peg Upright:** Reach → Grasp → Lift → Upright.

Each stage is associated with a well-defined geometric objective (e.g., approach, alignment, elevation, planar displacement, fine-grained goal adjustment). The following sections list the complete cost and potential functions for all stages used in STARE .

### D.1. Reach

**Goal.** Guide the end-effector toward the target object until it enters the vicinity where grasp alignment becomes feasible. This stage captures the coarse approach motion before any contact or fine-grained adjustment occurs.

**Cost.** We quantify reaching quality using the Euclidean distance between the end-effector position and the object's grasp point:

$$d_{\text{reach}}(t) = \left\| x^{\text{ee}}(t) - x^{\text{obj}}(t) \right\|.$$

The stage cost is the average approach error over the stage segment:

$$\ell_{\text{Reach}}(\tau^{(k)}) = \frac{1}{T_k} \sum_{t \in \tau^{(k)}} d_{\text{reach}}(t).$$

**Potential.** To provide dense shaping during the approach, we define a normalized potential that increases as the end-effector draws closer to the object:

$$\Phi_{\text{Reach}}(s) = \sigma\left( 1 - \frac{d_{\text{reach}}(s)}{d_{\text{reach}}^{\max}} \right).$$

Here $d_{\text{reach}}^{\max}$ is a geometry-derived normalization scale, chosen as the object's characteristic size:

$$d_{\text{reach}}^{\max} = L_{\text{obj}},$$

where $L_{\text{obj}}$ denotes the object's side length (e.g., cube size in ManiSkill3) or, more generally, its bounding-box diameter. The same $d_{\text{reach}}^{\max}$ is used as the threshold for the Reach $\to$ Grasp transition, ensuring that approach shaping and stage segmentation operate on a consistent geometric scale.

### D.2. Grasp

**Goal.** Establish and maintain a stable grasp. In ManiSkill environments, the end-effector approaches the object, aligns with the grasp point, and closes the gripper to securely hold the object.

**Cost.** The Grasp stage evaluates how well the end-effector pose matches the intended grasp configuration. We measure this using the distance between the tool-center point (TCP) and the object's grasp point:

$$d_{\text{pose}}(t) = \left\| x^{\text{tcp}}(t) - x^{\text{obj}}(t) \right\|.$$

The stage cost is the average geometric misalignment:

$$\ell_{\text{Grasp}}(\tau^{(k)}) = \frac{1}{T_k} \sum_{t \in \tau^{(k)}} d_{\text{pose}}(t).$$

**Potential.** We define a normalized potential that increases as the TCP approaches the grasp point:

$$\Phi_{\text{Grasp}}(s) = \sigma\left( 1 - \frac{d_{\text{pose}}(s)}{d_{\text{grasp}}} \right),$$

where $\sigma(\cdot)$ is the logistic sigmoid and $d_{\text{grasp}}$ is a rule-based normalization scale determined by object geometry:

$$d_{\text{grasp}} = L_{\text{obj}},$$

with $L_{\text{obj}}$ the characteristic object size (e.g., the cube side length in ManiSkill3). This scale is also used as the threshold for the Reach $\to$ Grasp stage transition, ensuring consistent geometry-aware shaping without per-task tuning.

### D.3. Transport

**Goal.** Move the grasped object toward its target goal pose while maintaining a stable grasp. This stage captures the coarse relocation of the object once it has been lifted or secured.

**Cost.** Transport quality is measured using the distance between the object's current position and the goal location:

$$d_{\text{trans}}(t) = \left\| x^{\text{obj}}(t) - x^{\text{goal}} \right\|.$$

The stage cost averages this residual goal error:

$$\ell_{\text{Transport}}(\tau^{(k)}) = \frac{1}{T_k} \sum_{t \in \tau^{(k)}} d_{\text{trans}}(t).$$

**Potential.** To provide dense shaping toward the goal, we define:

$$\Phi_{\text{Transport}}(s) = \sigma\left(1 - \frac{d_{\text{trans}}(s)}{d_{\text{trans}}^{\max}}\right).$$

The normalization scale $d_{\text{trans}}^{\max}$ is set to the characteristic task displacement:

$$d_{\text{trans}}^{\max} = \left\|x_{\text{init}}^{\text{obj}} - x^{\text{goal}}\right\|,$$

i.e., the distance between the object's initial position and its goal. This provides a rule-based scale reflecting the required transport distance, and aligns potential shaping with the physical extent of the manipulation.

### D.4. Place

**Goal.** Position the object precisely at the goal location and stabilize it. This stage captures the fine-grained alignment after coarse transport has moved the object near its target.

**Cost.** Placement quality is measured by the residual distance between the object's current position and the goal:

$$d_{\text{place}}(t) = \left\|x^{\text{obj}}(t) - x^{\text{goal}}\right\|.$$

The stage cost averages this near-goal deviation:

$$\ell_{\text{Place}}(\tau^{(k)}) = \frac{1}{T_k} \sum_{t \in \tau^{(k)}} d_{\text{place}}(t).$$

**Potential.** We define a normalized potential encouraging precise placement:

$$\Phi_{\text{Place}}(s) = \sigma\left(1 - \frac{d_{\text{place}}(s)}{d_{\text{place}}^{\max}}\right).$$

The normalization scale is chosen as the object's characteristic size:

$$d_{\text{place}}^{\max} = L_{\text{obj}},$$

which reflects the resolution required for fine positioning and provides a rule-based geometric scale.

### D.5. Push & Pull

**Goal.** Move the object toward the target goal while maintaining stable contact with the end-effector. This stage unifies push- and pull-based planar manipulation, as both correspond to contact-induced object translation in the plane.

**Cost.** Progress is measured using the residual distance between the object and its target:

$$d_{\text{pp}}(t) = \left\|x^{\text{obj}}(t) - x^{\text{goal}}\right\|.$$

The stage cost averages this residual error:

$$\ell_{\text{Push\&Pull}}(\tau^{(k)}) = \frac{1}{T_k} \sum_{t \in \tau^{(k)}} d_{\text{pp}}(t).$$

**Potential.** We define a normalized potential that increases as the object approaches the goal:

$$\Phi_{\text{Push\&Pull}}(s) = \sigma\left(1 - \frac{d_{\text{pp}}(s)}{d_{\text{pp}}^{\max}}\right),$$

where the normalization scale is chosen as the required planar displacement:

$$d_{\text{pp}}^{\max} = \left\|x_{\text{init}}^{\text{obj}} - x^{\text{goal}}\right\|.$$

This yields a rule-based geometric scale for shaping and is consistent with the Transport stage, differing only by the presence of contact.

### D.6. Lift

**Goal.** Lift the object from the table to a target height $z_{\text{goal}}$, ensuring that it is safely elevated above obstacles for subsequent manipulation.

**Cost.** We define the lift deviation as the vertical residual:

$$d_{\text{lift}}(t) = |z_{\text{obj}}(t) - z_{\text{goal}}|.$$

The stage cost averages this error:

$$\ell_{\text{Lift}}(\tau^{(k)}) = \frac{1}{T_k} \sum_{t \in \tau^{(k)}} d_{\text{lift}}(t).$$

**Potential.** We define a normalized shaping potential:

$$\Phi_{\text{Lift}}(s) = \sigma\left(1 - \frac{|z_{\text{obj}}(s) - z_{\text{goal}}|}{d_{\text{lift}}^{\max}}\right),$$

where

$$d_{\text{lift}}^{\max} = z_{\text{goal}} - z_{\text{table}}$$

is the required lifting displacement. This provides a rule-based geometric scale and yields dense shaping that increases smoothly as the object approaches its target height.

### D.7. Upright

**Goal.** Align the object's orientation with an upright target orientation. This stage captures fine-grained rotational adjustment after the object has been lifted or placed near its desired pose.

**Cost.** We measure uprightness using the angular deviation between the object orientation $R_{\text{obj}}(t)$ and the target upright orientation $R_{\text{upright}}$. Let

$$d_{\text{upright}}(t) = \arccos\left(\frac{\text{tr}\left(R_{\text{upright}}^{\top} R_{\text{obj}}(t)\right) - 1}{2}\right),$$

which equals the geodesic rotation distance on $\text{SO}(3)$. The stage cost averages this orientation error:

$$\ell_{\text{Upright}}(\tau^{(k)}) = \frac{1}{T_k} \sum_{t \in \tau^{(k)}} d_{\text{upright}}(t).$$

**Potential.** We define a normalized shaping potential that increases as the object approaches its upright orientation:

$$\Phi_{\text{Upright}}(s) = \sigma\left(1 - \frac{d_{\text{upright}}(s)}{d_{\text{upright}}^{\max}}\right).$$

The normalization scale $d_{\text{upright}}^{\max}$ corresponds to the maximum possible rotational deviation:

$$d_{\text{upright}}^{\max} = \pi,$$

which is the geodesic diameter of $\text{SO}(3)$. This choice yields a geometry-grounded scale that applies to any upright-orientation task.

### D.8. Goal (Push/Pull)

**Goal.** Ensure that the object reaches the target goal region and remains stably within it.

| Symbol | Meaning |
|--------|---------|
| $x^{\mathrm{ee}}(t)$ | End-effector (TCP) position at time $t$. |
| $x^{\mathrm{obj}}(t)$ | Object center or grasp-point position. |
| $x^{\mathrm{goal}}$ | Target goal position of the object. |
| $x^{\mathrm{obj}}_{\mathrm{init}}$ | Object position at episode start. |
| $d_{\mathrm{reach}}(t)$ | TCP–object distance during Reach. |
| $d_{\mathrm{pose}}(t)$ | TCP–object alignment error during Grasp. |
| $d_{\mathrm{trans}}(t)$ | Object–goal residual distance for Transport. |
| $d_{\mathrm{place}}(t)$ | Near-goal deviation during Place. |
| $d_{\mathrm{pp}}(t)$ | Residual distance for Push&Pull. |
| $d_{\mathrm{lift}}(t)$ | Vertical height error. |
| $d_{\mathrm{upright}}(t)$ | Geodesic rotation distance on SO(3). |
| $d_{\cdot}^{\mathrm{max}}$ | Normalization scales (stage-dependent). |
| $L_{\mathrm{obj}}$ | Characteristic object size (e.g., cube side length). |
| $T_k$ | Number of steps in stage $\tau^{(k)}$. |
| $R_{\mathrm{obj}}(t)$ | Object orientation at time $t$. |
| $R_{\mathrm{upright}}$ | Target upright orientation. |

*Table 3.* Summary of notation used across all stage definitions.

**Cost.** We measure goal attainment using the residual distance between the object and the goal:

$$d_{\mathrm{goal}}(t) = \left\| x^{\mathrm{obj}}(t) - x^{\mathrm{goal}} \right\|.$$

The stage cost averages this error:

$$\ell_{\mathrm{Goal}}(\tau^{(k)}) = \frac{1}{T_k} \sum_{t \in \tau^{(k)}} d_{\mathrm{goal}}(t).$$

**Potential.** A normalized potential provides dense shaping near the goal:

$$\Phi_{\mathrm{Goal}}(s) = \sigma\left( 1 - \frac{d_{\mathrm{goal}}(s)}{d_{\mathrm{goal}}^{\mathrm{max}}} \right),$$

where

$$d_{\mathrm{goal}}^{\mathrm{max}} = L_{\mathrm{obj}},$$

the characteristic object size. This creates a fine-scale potential landscape around the goal region, allowing smooth convergence without relying on a binary indicator.

**D.9. Notation Summary**

Table 3 lists all variables used in the stage-wise cost and potential definitions across all tasks.

# E. Extended OOD Experiment

We follow the protocol of (Liu et al., 2025) and construct three types of controlled distribution shifts on SimplerEnv. **(a) Vision OOD:** changes in background textures, lighting, and object appearance while preserving task semantics. **(b) Semantic OOD:** novel object categories, attributes, and language compositions that differ from the training instructions. **(c) Execution OOD:** variations in object initial poses, target locations, and contact geometries that alter the physical execution while keeping the goal unchanged.

**Results and Discussion.** Compared to trajectory-level baselines (SFT, GRAPE, RL4VLA), our STARE variants improve performance across vision, semantic, and execution OOD.

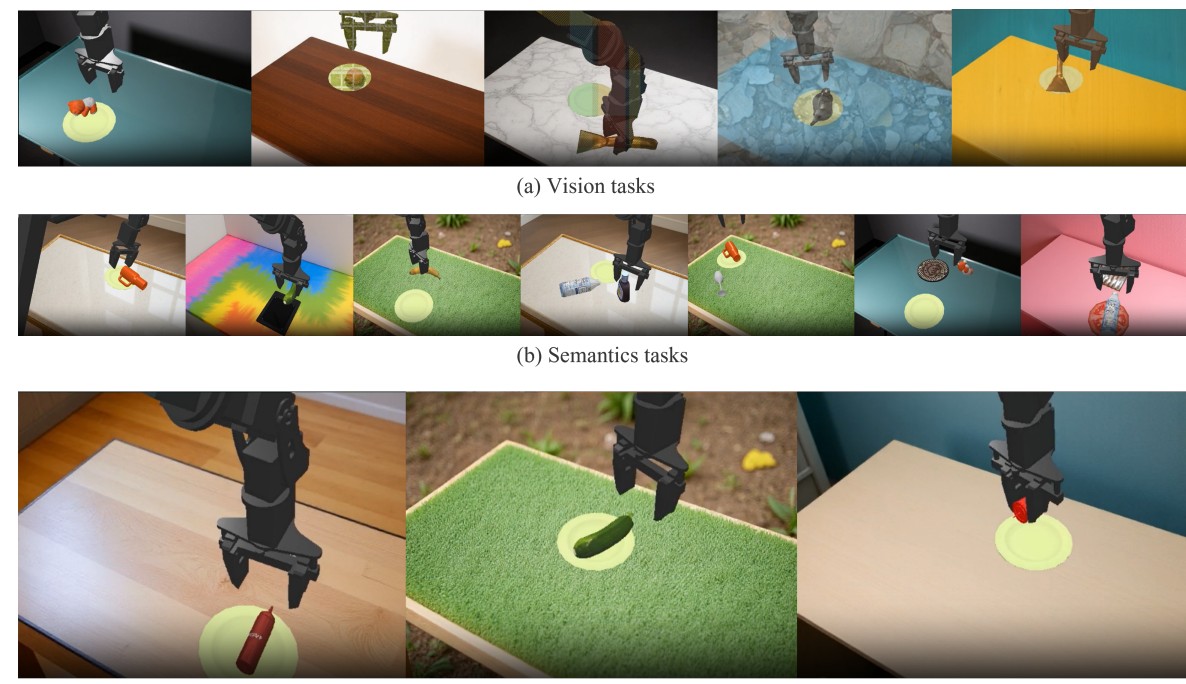

(a) Vision tasks

(b) Semantics tasks

(c) Execution tasks

*Figure 10.* Out-of-distribution (OOD) task variations.

*Table 4.* **OOD Generalization on SimplerEnv (OpenVLA-7B Backbone).** We report task success rates (%) under in-distribution (IND) and three types of out-of-distribution (OOD) shifts. All methods use the same OpenVLA-7B backbone and differ only in the fine-tuning strategy. The '(+X)' indicates the absolute improvement in success rate (%) over a relevant baseline.

| Model | Training (SimplerEnv-Bridge) | Vision (OOD) | Semantic (OOD) | Execution (OOD) | Avg. OOD |
|---|---|---|---|---|---|
| SFT | 41.7 | 27.3 | 28.7 | 30.3 | 28.8 |
| GRAPE | 43.9 | 31.3 | 30.7 | 33.7 | 31.9 |
| SFT → STA-TPO (STARE) | 51.6 | 39.0 | 37.3 | 40.7 | 39.0 (+7.1) |
| RL4VLA | 92.5 | 80.3 | 73.7 | 79.3 | 77.8 |
| SFT → STA-PPO (STARE) | 94.6 | 81.0 | **75.3** | **82.7** | 79.7 (+1.9) |
| IPI (STARE) | **98** | **93.3** | 68.7 | 78.0 | **80.0** |

STA-TPO improves the average OOD success rate by 7.1 over SFT, demonstrating the benefit of stage-aware preference alignment. STA-PPO further outperforms RL4VLA across all OOD settings, achieving the best overall trade-off between in-distribution and OOD performance. While IPI achieves the strongest in-distribution and vision OOD results, its semantic OOD score is slightly lower than STA-PPO. Nevertheless, IPI attains the highest average OOD performance, supporting our claim that the proposed pipeline improves generalization rather than overfitting.

## F. Related Work

**Long-horizon Robotic Manipulation Tasks in RL**    Long-horizon robotic manipulation involves completing a sequence of sub-tasks with frequent state and environment changes. Applying RL to such tasks is challenging due to sparse rewards, credit assignment, error accumulation, and high-dimensional state spaces. To address these issues, Plan-Seq-Learn (Dalal et al., 2024) leverages language models for high-level planning and RL for low-level control, enabling end-to-end execution from visual input to complex tasks. AC[3] (Yang et al., 2025) learns continuous action chunks with intrinsic rewards to mitigate sparsity. DEMO[3] (Escoriza et al., 2025) augments limited demonstrations with a world model and stage-wise dense rewards to improve sample efficiency. RoboHorizon (Chen et al., 2025d) employs LLMs to generate sub-goals and rewards, integrated with multi-view world models and planning to achieve high success rates. ARCH (Sun et al., 2025) combines high-level policy selection with a primitive skill library to tackle contact-rich assembly. PALM (Liu et al., 2026) improves long-horizon execution by leveraging affordance reasoning and within-subtask progress tracking. SARM (Chen et al.,

2025b) uses subtask-annotated reward modeling to filter demonstration quality, enabling robust long-horizon deformable object manipulation. RL-100 (Lei et al., 2025) emphasizes the importance of structured credit assignment and reliable objectives for long-horizon real-world robot learning. Building on these inspirations that divide goals into sub-goals, we take a further step with stage-aware reinforcement, decomposing trajectories into semantically meaningful stages. This provides denser feedback and enables progressive optimization, making RL more effective for VLA fine-tuning.

**RL Fine-tuning for LLMs** RL fine-tuning is a widely used approach for aligning LLMs. The most prominent method is RLHF (Ouyang et al., 2022), where a reward model trained from human preference data guides algorithms such as policy gradient or PPO to align outputs with human expectations. While highly successful, RLHF suffers from costly data collection and unstable training. To mitigate these issues, DPO (Rafailov et al., 2023) eliminates the reward model by directly optimizing on preference comparisons, simplifying training and improving stability. Further variants such as RLAIF (Lee et al., 2023) and RAFT (Dong et al., 2023) refine the framework. DeepSeek-R1 (Guo et al., 2025a) employs GRPO, which samples multiple responses per prompt and uses their relative performance within the group to compute advantages. As a subclass of LLMs, Large Reasoning Models (LRMs) (Zhang et al., 2025b) utilize Chain-of-Thought (CoT) (Wei et al., 2022) or Process Reward Model (PRM) (Lightman et al., 2023) for multi-step reasoning and face challenges akin to long-horizon VLA tasks, including sparse rewards and difficult credit assignment. This motivates our stage-aware reinforcement approach for VLA models.

**RL Fine-tuning for VLAs** Beyond recipe-style SFT that optimizes decoding and action representations for speed and success (Kim et al., 2025b). Recent studies explore RL as a fine-tuning paradigm for VLA models. GRAPE (Zhang et al., 2024) adapts DPO (Rafailov et al., 2023) to trajectory-level preferences to propose TPO, while ConRFT (Chen et al., 2025c) alternates RL and SFT in real-world settings. ReinboT (Zhang et al., 2025a) designs dense rewards, and (Guo et al., 2025b) propose an iterative SFT–RL pipeline to reduce instability and cost. RIPT-VLA (Tan et al., 2025) applies RLOO (Ahmadian et al., 2024) for online training, RL4VLA (Liu et al., 2025) studies RL-driven generalization, VLA-RL (Lu et al., 2025) applies PPO, and RFTF (Shu et al., 2025) introduces value models for dense reward estimation. SimpleVLA-RL (Li et al., 2025b) extends veRL to VLA models with GRPO-based online RL, demonstrating significant improvements in data efficiency. CO-RFT (Huang et al., 2025) proposes a chunked reinforcement learning framework for fine-tuning VLA models in real-world robotic tasks. VLA-RFT (Li et al., 2025a) proposes reinforcement fine-tuning of VLA policies using a learned world model to generate verified trajectory-level rewards, RLinf (Zang et al., 2025) provides a scalable and unified pipeline for VLA RL—combining rendering, inference, and training—to boost efficiency and performance. $\pi_{\mathrm{RL}}$ (Chen et al., 2025a) applied online RL fine-tuning for flow-based VLAs. $\pi_{0.6}^*$ (Intelligence et al., 2025) uses RECAP (Intelligence et al., 2025) to fine-tune VLAs with advantage-conditioned policies, learning from autonomous experience and expert corrections. GR-RL (Li et al., 2025c) combines offline data filtering with distributional critics and online latent-space RL for high-precision dexterous manipulation. Despite their promise, these methods typically optimize at the trajectory level, suffering from sparse rewards, coarse credit assignment, and difficult exploration in long-horizon manipulation. In contrast, our stage-aware RL decomposes trajectories into semantically meaningful stages and assigns stage-level rewards, providing denser, interpretable feedback and enabling progressive optimization for complex robotic tasks.

## G. Limitations and Future Directions

### G.1. Limitations

**Dependence on Hand-Crafted Rules.** The current implementation of STARE relies on manually designed geometric rules for stage decomposition. While these rules are based on well-established manipulation primitives, they require domain expertise to define and may not generalize seamlessly to tasks with fundamentally different manipulation structures. For instance, tasks involving flexible objects, liquids, or complex multi-object interactions may require significantly different stage definitions. Future work should investigate learned stage decomposition methods that can automatically discover meaningful task structure from data.

**Simulation-to-Reality Gap.** All experiments in this work are conducted in simulation environments. While these benchmarks provide valuable testbeds for algorithmic development, real-world deployment faces additional challenges including sensor noise, perception errors, actuation uncertainties, calibration drift, and physical wear. The stage boundaries detected reliably in simulation may become ambiguous in real settings where state observations are noisy or incomplete. Robust real-world deployment will require careful consideration of those factors, potentially including learned or adaptive stage detection mechanisms.

**Computational Requirements.** The proposed IPI pipeline requires more computational resources and training time compared to single-stage methods. While this investment yields substantial performance improvements, it may limit accessibility for researchers with constrained computational budgets. Additionally, the stage-aware reward calculation introduces modest computational overhead during online training, though this is typically negligible compared to policy network inference.

### G.2. Future Directions

To address the limitations and risks identified above, we propose several important directions for future work:

- **Learned Stage Decomposition:** Developing methods to automatically discover meaningful task structure from data, reducing dependence on hand-crafted rules.

- **Real-World Validation:** Systematic evaluation on physical robots with careful analysis of sim-to-real transfer and failure modes.

- **Human-in-the-Loop:** Integrating human feedback more tightly into the stage-aware training pipeline, particularly for safety-critical stages.

- **Broader Task Coverage:** Extending stage-aware methods to more challenging benchmarks and more diverse manipulation domains including bi-manual manipulation, dexterous in-hand manipulation and mobile manipulation.

