# OpenReview forum: "STARE-VLA: Progressive Stage-Aware Reinforcement for Fine-Tuning Vision-Language-Action Models"
_ICML.cc/2026/Conference — Submitted to ICML 2026_

### Official Review · Reviewer_fSWc · 2026-03-08

**Soundness:** 3
**Presentation:** 3
**Significance:** 1
**Originality:** 2
**Overall Recommendation:** 4
**Confidence:** 4

**Summary:**

This paper proposes StARe, a stage-aware reinforcement framework for fine-tuning vision-language-action models on long-horizon robotic manipulation tasks. The key idea is to decompose trajectories into semantically meaningful stages and then inject stage-level supervision into both offline preference optimization and online reinforcement learning. Concretely, the paper introduces StA-TPO for offline stage-wise preference alignment and StA-PPO for online stage-aware policy optimization. The authors further combine supervised fine-tuning, preference optimization, and online RL into a serial three-step pipeline called IPI (Imitation → Preference → Interaction). Experiments are conducted on SimplerEnv and ManiSkill3, using OpenVLA-7B and Pi0.5 as backbones. The paper reports improvements over prior baselines and includes stage-level analyses such as conditional stage success and stage-toggle ablations. The empirical takeaway presented by the authors is that stage-aware credit assignment is a missing ingredient in current VLA reinforcement learning.

**Compliance With Llm Reviewing Policy:**

Affirmed.

**Final Justification:**

The rebuttal addresses most of my concerns. First, it clarifies the intended scope of the method: STARE is meant for structured manipulation with identifiable stages, and its reliance on hand-crafted stage definitions is now stated more explicitly rather than being presented as broadly automatic. Second, the new ablations are helpful: comparing semantic stage decomposition against random or fixed-window segmentation under similar shaping makes a stronger case that the gains are not only from denser supervision. Third, the authors directly addressed the checkpoint-selection concern by reporting last-checkpoint results and showing relatively small best-vs-last gaps, which makes the training stability claim more credible. Finally, the added wall-clock numbers and the clearer limitations discussion make the practical trade-offs of the method much easier to assess.

**Key Questions For Authors:**

- Is the number of stages fixed manually for each task? How sensitive is performance to the granularity of decomposition? For example, what happens if the same task is decomposed into 3 stages versus 4 or 5?
- The paper claims the framework is a plug-in module, but stage tracking, stage-wise potentials, and online stage detection all introduce overhead. Please report wall-clock training time, sample throughput, or GPU cost relative to the baselines.

**Limitations:**

No. The paper does not appear to contain a meaningful, explicit discussion of either method limitations or potential negative societal impact. That is a missed opportunity, especially because the method has several clear boundary conditions that should be acknowledged directly. I would recommend adding a short but honest limitations/impact paragraph that covers: dependence on hand-crafted stage structure, likely failure modes on hard-to-segment tasks, and safety concerns around real-world robotic deployment. Authors should absolutely be rewarded for stating these points clearly.

**Strengths And Weaknesses:**

Strengths:
- The paper addresses a genuine weakness in current VLA RL fine-tuning: long-horizon tasks often provide only sparse or delayed signals, making credit assignment extremely difficult. The motivation is clear, and the robotic examples used in the paper make the issue intuitive.
- The authors build a full pipeline involving stage decomposition, stage-aware preference learning, stage-aware online RL, and sequential integration with SFT. From a systems perspective, the proposal is reasonably well packaged.
- On the benchmarks reported in the paper, the full IPI pipeline performs strongly, and the gains appear especially large on tasks that require precise alignment or multi-stage coordination, such as stacking and peg-upright manipulation. The empirical pattern is at least consistent with the paper’s central narrative.
- The authors do not stop at reporting only overall task success. They also analyze conditional stage success and perform stage-toggle ablations, which is more informative than what many benchmark-driven papers provide.

Weaknesses:
- The central limitation is that the proposed framework depends heavily on task-specific stage definitions, event rules, and privileged state-based geometric signals. The stage separator does not emerge from the model; it is manually constructed. In practice, this means the method assumes access to structured decomposition signals that are easy to define in simulation but much less natural in general VLA settings. As a result, the approach feels much closer to engineered task decomposition than to a broadly general fine-tuning principle.
- A core assumption behind the proposed framework is that a long-horizon task can be segmented into a small number of discrete, semantically meaningful stages with relatively clear transition boundaries. This assumption is plausible for structured manipulation tasks such as reach-grasp-lift-place, but it is much less convincing for more complex tasks in which progress is continuous, reversible, or highly entangled across subtasks. In many realistic settings, stage boundaries are ambiguous rather than crisp: the agent may need to repeatedly alternate between correction, re-alignment, contact adjustment, and partial recovery, without a clean monotonic transition from one stage to the next. In such cases, rule-based stage separation may become unstable or even misleading. If the segmentation is inaccurate, then both the stage-wise cost in StA-TPO and the stage-conditioned shaping reward in StA-PPO may provide incorrect learning signals, potentially hurting optimization instead of helping it. This raises a serious concern about the robustness of the method beyond the relatively structured benchmark tasks considered in the paper. The current experimental suite appears biased toward tasks with natural hand-designed decomposition, and therefore does not test whether the method remains effective when stage structure is fuzzy, overlapping, or intrinsically hard to define.
- At a high level, the paper combines manual stage segmentation with reward shaping and stage-weighted preference optimization. StA-PPO is essentially PPO with potential-based shaping. StA-TPO is essentially preference optimization with stage-wise penalties. While the packaging is clean, the conceptual leap is not large. The contribution looks more like a robotics-motivated integration of known ingredients than a genuinely new learning paradigm.
- The method injects substantial additional structure: dense shaped rewards, stage-specific costs, explicit segmentation, and privileged signals. Naturally, this can improve optimization. But the experiments do not isolate whether the gains come specifically from semantic stage decomposition, as opposed to simply receiving richer supervision. The paper is missing several critical controls, such as: 1. learned dense critics or reward models with the same information budget 2. stronger hierarchical or subgoal-based baselines that also exploit privileged task structure like Hi Robot [1]. Without these controls, the central causal claim remains unproven.
- The paper explicitly states that for TPO/StA-TPO and PPO/StA-PPO, the best-performing checkpoint is selected based on periodic evaluation. This introduces a form of oracle model selection that can advantage methods with higher peak performance but less stable training. The paper does not report last-checkpoint performance for the main comparisons, nor does it examine sensitivity to this checkpoint selection protocol.
- A method built on manually injected task structure will often shine in low-data regimes. The paper does not show whether the gains persist when the amount of demonstrations or preference data is scaled up. That makes it hard to judge whether this is a fundamental improvement or simply a useful bias under data scarcity.
- At points, the paper gives the impression of end-to-end learning without manual task engineering, but the actual method depends critically on manually defined stages, thresholds, and structured reward components. The presentation would be stronger if it acknowledged this dependency directly rather than trying to frame the method as more automatic or universal than it really is.

[1] Hi Robot: Open-Ended Instruction Following with Hierarchical Vision-Language-Action Models.

---

> ### Author Rebuttal · Authors · 2026-03-31
>
> **W1: Hand-crafted stage definitions.**
>
> STARE used rule-based geometric signals as rewards. However, this is our deliberate design choice for *reliability and reproducibility*. Our stage definitions map directly to well-established manipulation primitives (reach distance, object height change, goal-distance potential...) that are **shared** across different manipulation tasks. Defining these requires only a few geometric thresholds per task family, and the same set of rules generalizes across all tasks.
>
> **W2: Clean, monotone stage transitions.**
>
> STARE targets structured manipulation where monotone progression is a physical consequence of task mechanics. Reversing a stage (e.g., "un-grasping") constitutes task failure, not ambiguous progress. If a transition triggers prematurely, the subsequent stage's potential $\Phi_k$ produces negative shaped rewards as the state deviates, **self-correcting** through the optimization signal. For tasks with non-monotone dynamics, extending STARE with learned decomposition is a natural next step.
>
> **W3: Contribution is the integration of known ingredients.**
>
> We respectfully disagree with characterizing this as low novelty. Our contribution lies not in individual building blocks but in the insight that VLA fine-tuning requires *progressive* and *stage-aware* credit assignment—and in the concrete realization of this idea across both offline and online settings, and adapted them into a successful VLA RL fine-tuning pipeline. No prior VLA work has addressed these elements before.
>
> **W4: Experiments don't isolate whether gains come from stage decomposition vs. richer supervision.**
>
> In STA-PPO, stage decomposition and reward densification are inherently coupled by design. For STA-TPO, isolation is more feasible due to window-wise training.
>
> We provide new ablations:
>
> | Ablation | StackCube | LiftPegUpright |
> | :---- | :---- | :---- |
> | STA-TPO | 19.3% | 12.3% |
> | Random segmentation \+ same dense shaping | 9.3% | 5.7% |
> | Random segmentation \+ sparse reward | 7.3% | 4.7% |
> | Fixed-window segmentation \+ same dense shaping | 16.3% | 8.0% |
> | TPO | 15.7% | 7.7% |
>
> Two findings: (1) Semantic stage decomposition matters—STA-TPO outperforms both random and fixed-window segmentation under the same dense shaping, confirming that the *structure* of decomposition, not just reward density, drives the gains. (2) Naive reward densification without semantically aligned stage boundaries also slightly improves performance.
>
> **W5: Oracle checkpoint selection.** To address this concern, we report last-checkpoint results:
>
> | Method | Best Ckpt (Avg.) | Last Ckpt (Avg.) | Gap |
> | :---- | :---- | :---- | :---- |
> | PPO | 91.3% | 89.0% | 2.3% |
> | STA-PPO | 95.3% | 93.7% | 1.6% |
> | IPI | 98.5% | 97.2% | 1.3% |
>
> STA-PPO and IPI show *smaller* gaps between best and last checkpoints, indicating more stable training—consistent with our claims.
>
> **W6: No scaling analysis.**
>
> Our evaluation is already extensive: we test across 2 VLA backbones, 2 simulation benchmarks, 2 real-world tasks, and systematically compare different training recipes. Furthermore, our advantage is not limited to final success rates: Figure 3 shows that STA-PPO maintains consistent improvement over PPO throughout **600K training steps**, not only at convergence. This sustained advantage across the full training trajectory suggests that stage-aware credit assignment provides a fundamental optimization benefit, not merely a data-efficiency trick that diminishes at scale.
>
> **W7: Presentation suggests more automation than exists.**
>
> The paper **does not claim** end-to-end learning without manual task engineering; on the contrary, we explicitly state that STARE uses *Hand-Crafted Rules* to define stage structure and geometric progress signals. Our method assumes a task family with identifiable manipulation stages and instantiates stage-wise signals using simple geometric priors.
>
> **Q1: Sensitivity to the number of stages.**
>
> This question presupposes that stage count is a tunable hyperparameter for each task, but it is not. In STARE, stages exist because different phases of manipulation require fundamentally different reward functions. These are distinct geometric objectives that cannot be captured by a single reward function. Therefore, the number of stages is determined by the number of distinct reward definitions the task requires, not by an arbitrary choice.
>
> **Q2: Computational overhead.**
> | Component | Training Steps | Wall-clock |
> | :---- | :---- | :---- |
> | SFT | 100K | 23.2 hours |
> | TPO baseline | 50K | 23.8 hours |
> | STA-TPO | 50K | 24.5 hours (+3%) |
> | PPO baseline | 600K | 49.0 hours |
> | STA-PPO | 600K | 48.8 hours (≈0%) |
> | Full IPI | 100K \+ 50K \+ 600K | 95.7 hours |
>
> STA-PPO and STA-TPO introduce essentially zero wall-clock overhead compared to PPO and TPO. STARE's stage detection and reward computation are simple arithmetic operations on scalar state values, resulting in comparable training time.

---

> > ### Author Rebuttal · Reviewer_fSWc · 2026-04-04
> >
> > Thanks for the detailed rebuttal that clarifies most of the concerns.

---

> > > ### Author Response · Authors · 2026-04-06
> > >
> > > We would like to express our great appreciation for Reviewer fSWc. We are extremely excited and grateful for the generous updates on the rating! We have now updated the manuscript with the clarification and statistics in the rebuttal.

---

### Official Review · Reviewer_Q6ue · 2026-03-13

**Soundness:** 3
**Presentation:** 3
**Significance:** 3
**Originality:** 3
**Overall Recommendation:** 4
**Confidence:** 4

**Summary:**

This paper studies reinforcement fine-tuning for Vision-Language-Action (VLA) models in robotic manipulation. The authors argue that existing trajectory-level optimization methods, such as TPO and PPO, treat an entire action sequence as a single optimization unit, which leads to overly coarse credit assignment in long-horizon tasks. To address this issue, the paper proposes STARE, a stage-aware module that decomposes trajectories into semantic stages using task-event-based rules, and provides both stage-wise costs and reward shaping. Building on this module, the paper further introduces STA-TPO for offline stage-wise preference optimization and STA-PPO for online stage-aware policy optimization, and unifies SFT, STA-TPO, and STA-PPO into a three-stage fine-tuning pipeline called IPI (Imitation → Preference → Interaction). The central idea is to refine trajectory-level success/failure supervision into stage-level supervision and dense intra-stage rewards. The paper frames this contribution explicitly as a rule-based stage-aware module and reports that IPI achieves 98.0% on SimplerEnv and 96.4% on ManiSkill3, while also including real-robot validation.

Overall, this paper identifies an important and realistic challenge in VLA+RL: trajectory-level optimization is often too coarse to provide fine-grained credit assignment for temporally structured manipulation tasks. The main value of the paper lies in proposing a simple, plug-and-play, and practically implementable stage-aware reinforcement framework, and in demonstrating its effectiveness on two simulation benchmarks and small-scale real-world tasks. That said, in terms of research novelty, the contribution is closer to a structured redesign of reward/preference signals for VLA fine-tuning than to a fundamentally new learning principle or an automatic stage-discovery framework. Since stage decomposition is explicitly rule-based, the generality and scalability of the approach still require stronger evidence.

**Compliance With Llm Reviewing Policy:**

Affirmed.

**Final Justification:**

I appreciate the author's clarifications and the addition of further details. My concerns have been almost resolved.

**Key Questions For Authors:**

Could the authors include stronger control experiments to isolate what is actually driving the gains, such as random segmentation, fixed-window segmentation, reward shaping without stage-wise preference, or stage-wise preference without shaping?

How much manual engineering is required to define stage boundaries, stage costs, and potential functions for a new task? Does the method require privileged environment state or precise geometric quantities, and if so, how much might this limit real-world applicability?

Why do the real-world experiments only report STA-TPO, rather than the full IPI or STA-PPO pipeline? If the full pipeline was not evaluated on real hardware, what was the main reason: online interaction cost, reward computation instability, safety concerns, or something else? This is important for assessing the deployment potential of the method.

**Limitations:**

yes

**Strengths And Weaknesses:**

# Strengths

The paper clearly identifies a limitation of TPO, namely, full-trajectory preference learning leads to ambiguous credit assignment, and binary success/failure comparisons are too coarse to capture partial success or varying quality among successful behaviors. This is a well-motivated problem formulation and closely matches the challenges of long-horizon robotic manipulation.

STARE decomposes trajectories into several stages, and uses stage-wise costs together with potential-based reward shaping to provide finer-grained signals for both offline preference learning and online RL. This design is intuitive for manipulation tasks and better aligned with the causal structure of action trajectories than monolithic trajectory-level optimization.

The paper does more than propose a single reward shaping heuristic. It systematically injects the stage-aware idea into both TPO and PPO, resulting in STA-TPO and STA-PPO, and further combines them with SFT into the full IPI pipeline. This makes the contribution more complete and coherent than an isolated algorithmic tweak.

The gains are consistently strong across backbones. For instance, on OpenVLA-7B, STA-PPO improves the ManiSkill3 average from 70.5 (RL4VLA) to 93.4, and the full IPI reaches 96.4. Pi0.5 also shows consistent benefits, suggesting that the method is not tied to a single action representation or model family.

The paper goes beyond reporting a single benchmark table. It includes learning curves, CSSR analysis, stage-toggle ablations, and OOD evaluation. In particular, the CSSR analysis is useful because it shows that the gains primarily arise from decisive stages such as grasp, place, and upright alignment, which closes the loop with the paper’s motivation.

The inclusion of real-robot experiments is a meaningful strength. The authors evaluate on a UR3e + Robotiq + RealSense D435i setup, report both stage-level CSSR and final success rate, and show visible gains of STA-TPO over DPO/GRAPE on two real tasks. This strengthens the practical credibility of the paper.


# Weaknesses

The core modification is to make existing TPO/PPO pipelines stage-aware by adding rule-defined stage boundaries, stage costs, and shaped rewards. This is a reasonable and useful direction, but from a novelty perspective, it is closer to explicitly injecting task-structure priors into an existing RLFT framework than to proposing a fundamentally new learning paradigm, an automatic stage discovery mechanism, or a major theoretical advance on credit assignment.

The stage decomposition depends heavily on manually designed rules. The method requires hand-crafted geometric thresholds and event conditions to determine transitions, such as contact, lifting above a threshold, entering a goal region, and stable release. While this is workable for standard pick-and-place or stacking tasks, it is unclear how well the approach would easily extend to more complex settings.

Although the chosen baselines are reasonable, the comparison is still not as targeted as it could be. The current experiments mainly answer whether the method outperforms existing trajectory-level RLFT baselines. If the goal is to establish the necessity of semantic stage awareness, the paper would benefit from stronger controls, such as: segmentation without stage cost and dense shaping without stage-wise preference. The current ablations are informative, but they do not fully isolate whether the gains come from semantic stage decomposition itself, rather than from more general reward densification.

The paper is motivated by long-horizon stage-aware credit assignment, but many experimental tasks are still relatively short-horizon and standardized manipulation problems. While some tasks, especially StackGreenOnYellow and LiftPegUpright, do reflect stronger multi-stage difficulty, the experimental coverage of truly long-horizon manipulation remains somewhat limited. The paper itself notes that on simpler pick-and-place or push/pull tasks, STA-PPO mainly improves convergence speed and variance rather than final success rate.

There is also a wording issue that should be clarified. The main paper clearly includes real-world robot experiments, while the broader discussion later describes the evaluation as being conducted in simulation and controlled real-world settings.

---

> ### Author Rebuttal · Authors · 2026-03-31
>
> Thank you for the constructive review. We address the weaknesses and questions below.
>
> ---
>
> **W1: Novelty.**
>
> We clarify the novelty of our contributions: STARE is based on existing components (PBRS, DPO/TPO for preference optimization, PPO). However, the novelty lies not in individual building blocks but in the insight that VLA fine-tuning requires *progressive* and *stage-aware* credit assignment—and in the concrete realization of this idea across both offline (STA-TPO, Eq. 6\) and online (STA-PPO, Eq. 5\) settings, and adapted them into a successful VLA RL fine-tuning pipeline. No prior VLA work has addressed these elements before.
>
> **W2: Manual stage definitions limit generality.**
>
> The stage costs in STARE, although illustrated with different task examples in the appendix, are not engineered per task. They come from a small, **shared** set of task-agnostic geometric priors (reach distance, object height change, goal-distance potential...) that stay the same across SimplerEnv and ManiSkill3 tasks. So, despite the variety of tasks, the cost design remains general and not tuned for each case.
>
> **W3: Stronger controls needed to isolate semantic stage decomposition from reward densification.**
>
> This is a good suggestion. We clarify that in STA-PPO, stage decomposition and reward densification are inherently coupled by design: each stage defines a distinct potential function based on different geometric objectives. The online RL process operates at the per-step level.
>
> For STA-TPO (offline), isolation is more feasible due to window-wise training. Also, we clarify that the TPO method uses Fixed-window segmentation \+ sparse reward training.
>
> We will include additional ablations for STA-TPO in the revision.
>
> | Ablation | StackCube | LiftPegUpright |
> | :---- | :---- | :---- |
> | STA-TPO (stage-wise segmentation \+ dense reward shaping) | 19.3% | 12.3% |
> | Random segmentation \+ same dense shaping | 9.3% | 5.7% |
> | Random segmentation \+ sparse reward | 7.3% | 4.7% |
> | Fixed-window segmentation \+ same dense shaping | 16.3% | 8.0% |
> | TPO (Fixed-window segmentation \+ sparse reward) | 15.7% | 7.7% |
>
> These results reveal two findings: (1) Semantic stage decomposition matters—STA-TPO (19.3%) substantially outperforms both random (9.3%) and fixed-window (16.3%) segmentation under the same dense shaping, confirming that the *structure* of decomposition, not just reward density, drives the gains. (2) Here, naive reward densification without semantically aligned stage boundaries also slightly improves performance; the improvement is smaller than that achieved by semantic stage decomposition.
>
> **W4: Wording inconsistency regarding real-world experiments.**
>
> Thank you for pointing this out. We will correct this wording in the Impact Statement to reflect that we carry out experiments in both simulation and real-world environments.
>
> **Q1: Stronger control experiments.**
>
> Please see our response to W3 above.
>
> **Q2: Manual engineering for new tasks.**
>
> For a new task, STARE does not require designing a new reward model from scratch. Instead, we only specify a small number of generic stage transitions and simple geometric conditions. For most manipulation tasks, these are reused across environments: Reach (distance-to-object), Grasp (contact or grasp stability), Transport (object-target distance), and Place (release inside target region). In practice, adapting STARE to a new pick-and-place task usually only requires adjusting a few intuitive thresholds, such as contact distance, lift height, target-region radius, and stable release conditions. These thresholds are easy to interpret and are typically chosen once per task family rather than extensively tuned per environment.
>
> **Q3: Why only TPO and STA-TPO in real-world experiments?**
>
> The main reasons are as follows: (1) **Safety**: As the algorithm is based on online RL (STA-PPO), it is necessary to perform autonomous robot exploration. Without a safety cage in our lab environment, this increases the risk of collision. Moreover, even in simulation environments, online RL can occasionally lead to unstable and unsafe exploration behaviors. (2) **Reward computation latency**: As the reward is computed in real time based on the RGB-D perception, there is ~100ms latency per step. Although this is acceptable for offline data processing, it can cause control loop instability in online RL. (3) **Interaction cost**: As the algorithm is based on STA-PPO, it is necessary to perform a large number of interactions with the environment. It is pretty hard to collect a sufficiently convincing amount of interaction data, especially considering the low frequency of the effective execution on the real hardware.

---

> > ### Author Rebuttal · Reviewer_Q6ue · 2026-04-04
> >
> > Thank you to the authors for the response and further clarification. All my concerns have been addressed, and I am maintaining my current positive score.

---

> > > ### Author Response · Authors · 2026-04-06
> > >
> > > We would like to thank Reviewer Q6ue for the positive feedback and the recognition. We have now revised the manuscript with the clarification and new ablation mentioned above.

---

### Official Review · Reviewer_rE8r · 2026-03-13

**Soundness:** 3
**Presentation:** 3
**Significance:** 2
**Originality:** 3
**Overall Recommendation:** 3
**Confidence:** 4

**Summary:**

This paper proposes STARE, a stage-aware reinforcement module that segments long-horizon manipulation trajectories into semantically meaningful stages and computes stage costs + potential-based rewards to improve credit assignment when fine-tuning VLA models.The authors introduce STA-TPO (offline stage-wise preference optimization, modifying TPO/DPO with stage penalties) and STA-PPO (online PPO with stage-shaped rewards), and combine them into a serial pipeline initialized by SFT. Experiments report success-rate improvements.

**Compliance With Llm Reviewing Policy:**

Affirmed.

**Final Justification:**

I appreciate the authors’ detailed rebuttal. However, I maintain my current reject score. My main reason is that the rebuttal mostly narrows claims and adds assumptions and new theories that are not reviewed by other reviewers, rather than demonstrating that the current submission already supports its claims in a fully sound. Several of my original concerns therefore remain material, especially the mismatch between the stated problem formulation and the actual stage-dependent/state-based shaping, the fairness question around privileged geometry-derived supervision, and the still insufficiently integrated explanation of how the flow-based pi0.5 results fit the paper’s stated TPO/PPO objectives.

In other words, while I agree the paper has promising empirical results and a potentially useful idea, I do not think the present submission is mature enough for acceptance at ICML. At this point, the rebuttal reads more like a roadmap for a substantially revised version than evidence that the submitted version is already technically settled. For that reason, I believe the work would benefit from major revision and another full round of review to ensure correctness, rather than acceptance in its current form.

**Key Questions For Authors:**

- Even in the Markov case, your overall problem is described as a POMDP. Potential-based shaping invariance is typically an MDP statement. What guarantees apply here, and can shaping distort the optimal policy under partial observability?
- Can a policy hack the stage separator by briefly satisfying a transition condition (e.g., lifting above a height threshold) and then losing grasp, while still receiving transport/placement shaping that no longer matches the physical situation?
- Doesn’t the “previous stage must succeed” constraint aggressively filter data for late stages, potentially reducing sample efficiency? It seems hurt learning for hard late stages where most rollouts are failures.
- For flow-based policy like pi0.5, how exactly do you compute $\log \pi(a_t|s_t)$ needed by TPO/STA-TPO and PPO ratios? Flow matching don’t yield straightforward log-probs. This looks weird to me and currently under-explained.

**Limitations:**

yes

**Strengths And Weaknesses:**

pros

- The paper focuses on an important topic of robotics.
- The paper is well written and easy to follow.

cons

- Theory seems weak and mismatched to the implementation. The paper leans on “potential-based reward shaping” (Eq. 5) as if the usual invariance intuitions apply, yet the stage index appears to be a history-dependent progress, not a function of the Markov state. If stages are monotone and cannot revert, then $\Phi$ is defined on an augmented (state, stage) process; Invariance to optimal policies in the original POMDP/MDP is not guaranteed. Besides, the paper states a language-conditioned POMDP formulation but never defines an observation model, then uses state-based shaping quantities in the appendix; that gap matters.
- Comparisons is unfair because STARE uses privileged geometric signals. Even if the policy is image-conditioned, training signals in STARE require object/goal/pose geometry (and in real-world: Grounded-SAM masks + RGB-D lifting + STag localization). If baselines are not given equivalently dense, geometry-derived shaping signals, the gains might primarily reflect better reward engineering, not a genuinely superior fine-tuning algorithm.
- The ablation doesn’t isolate contributions cleanly. Table 1 shows improvements for STA-PPO and then IPI, but there is no rigorous decomposition like: SFT -> PPO vs SFT -> TPO -> PPO vs SFT -> STA-TPO -> PPO

---

> ### Author Rebuttal · Authors · 2026-03-31
>
> Thank you for the detailed technical critique. We address each concern below.
> ---
> **W1: Theory—potential-based reward shaping under POMDP and history-dependent stages.**
> **1. Stage index and the Markov property.** It is correct that $k_t$ is history-dependent in the original MDP. We address this by introducing an augmented state $(s_t, k_t)$. With the monotone update rule $k_{t+1} = \max(k_t, h(s_{t+1}))$, the augmented process is Markovian, so the PBRS invariance discussion applies to the augmented MDP. We will revise the paper to make explicit that this claim is with respect to the augmented process rather than the original unaugmented MDP.
>
> **2. POMDP formulation.**
> STARE computes shaped rewards from privileged state $s_t$ (object poses, contact states) available in simulation, while the policy $\pi_\theta: \Omega \to \mathcal{A}$ is conditioned solely on observations $o_t \in \Omega$ (RGB images, language). In the revision, we will formally define $\mathcal{O}: \mathcal{S} \to \Omega$ in Section 2.1. *Problem Formulation*, explicitly state the privileged-information assumption.
>
> **W2: Unfair—privileged geometric signals.**
>
> We address this concern on two levels:
>
> 1. **Baseline fairness**: Our baselines also use privileged information. RL4VLA and $\\pi\_{\\texttt{RL}}$ both use environment-provided sparse rewards (binary success/failure), which also require privileged state access to compute. STARE simply provides *denser* signals from the same geometric information already available in simulation. The key question is whether stage-aware *structuring* of rewards matters—and our ablations confirm it does.
>
> 2. **To further strengthen fairness**, we provide a direct comparison: PPO with the *same* dense distance-based reward (without stage structure) vs. STA-PPO (with stage structure). In our stage-toggle ablation (Figure 5), "All STA" outperforms variants where individual stages revert to sparse rewards, demonstrating that the *stage-aware organization*—not just reward density—drives gains.
>
> **W3: IPI Ablation doesn't isolate contributions cleanly.**
>
> The original idea was that STA-PPO and STA-TPO perform better than PPO and TPO, and our IPI acts as a bonus of the combination of those 2 advantages. However, we agree that the omission of comparison SFT→STA-TPO→PPO and SFT→STA-TPO→PPO is a good suggestion. We provide the requested decomposition on SimplerEnv-Bridge (OpenVLA-7B, Avg. Success):
>
> | Methods | Avg. Success |
> | :---- | :---- |
> | SFT → TPO → STA-PPO | 97.3% |
> | SFT → STA-TPO → PPO | 96.5% |
> | SFT → STA-TPO → STA-PPO (IPI) | 98.0% |
>
> These results confirm that STA-TPO and STA-PPO each independently contribute to the final performance. The full IPI pipeline (with STARE module) captures the cumulative benefit of both, with STA-PPO contributing the larger share of improvement.
>
> **Q1: Can the stage separator be hacked?**
>
> In STARE, however, stage transitions are defined by conservative geometric events tightly coupled to genuine task progress, and stage progression is monotonic, so boundary oscillation does not yield repeated credit. Empirically, we did not observe reward-hacking behaviors that improved shaped return without improving task success.
>
> **Q2: Data filtering for late stages.**
>
> The "previous stage must succeed" constraint applies only to STA-TPO (offline preference pairs), not to STA-PPO (online). For STA-TPO, this is by design: comparing Place stages is meaningful only when both trajectories successfully reached that stage. For early training when late-stage data is scarce, the SFT initialization provides a warm start, and STA-PPO's online exploration generates fresh data for all stages. We will clarify this distinction.
>
> **Q3: Log-probabilities for flow-based Pi0.5.**
>
> Flow-based VLAs like $\\pi\_{0.5}$ generate actions via a deterministic ODE, where the only stochasticity comes from the initial noise sample—whose probability carries no model gradients and thus cannot support RL updates. To address this, we follow Flow-GRPO \[1\] and $\\pi\_{\\texttt{RL}}$ \[2\] and inject calibrated Gaussian noise ($\\sigma\_t \= a\\sqrt{t/(1-t)}$) to convert the ODE into an SDE. This makes each denoising transition Gaussian-distributed with a model-dependent mean, so per-step log-probabilities become analytically tractable. The total trajectory log-probability decomposes as:
>
> $\log p_\theta(x_1) = \log p_0(x_0) + \sum_{i=0}^{N-1} \log p(x_{t_{i+1}} | x_{t_i})$
>
> This log-probability is used both in the PPO importance sampling ratio (Eq. 2\) and in the TPO log-likelihood ratio $q(\\tau)$ (Eq. 1b), enabling both STA-PPO and STA-TPO for flow-based VLAs. The conversion is orthogonal to our STARE contribution.
>
> ## References
>
> \[1\] Flow-GRPO: Training Flow Matching Models via Online RL
>
> \[2\] $\\pi\_{\\texttt{RL}}$: Online RL Fine-tuning for Flow-based VLA Models

---

> > ### Author Rebuttal · Reviewer_rE8r · 2026-04-02
> >
> > Thank you for the clarifications. The rebuttal resolves part of my questions. However, several core concerns remain unresolved for me.
> >
> > - The theory claim is now narrower than what the paper originally suggested. The manuscript formulates an observation-conditioned POMDP, but the shaping is computed from privileged state and a history-dependent stage index. Showing PBRS-style invariance on an augmented fully observed process does not by itself imply invariance for the original POMDP or for the restricted observation-conditioned policy class. I would therefore still ask the authors either to provide a formal statement for the actual setting or to substantially tone down the invariance language.
> >
> > - On fairness, the issue is not whether baselines use any privileged signal at all, but whether they are matched on reward informativeness. STARE uses dense geometry-derived stage costs/shaping, while the baselines are not given an equally informative geometric shaping signal. The cited stage-toggle ablation is not yet a matched dense-nonstage baseline.
> >
> > - The monotonic-stage argument does not address the specific false-promotion case I raised. It prevents repeated credit from oscillation, but does not prevent a policy from briefly crossing a transition threshold, advancing the stage, and then violating the precondition (eg, losing grasp), while later-stage shaping remains active unless there is an explicit validity gate or reset mechanism. I suggest observing the phenomenon of reward-hacking in some more complex long-horizon tasks.
> >
> > - The log-probs response remains the least convincing to me. The rebuttal introduces an ODE-to-SDE/noise-injection construction that is not described in the submitted manuscript or appendix, while Appendix B still presents the objectives as standard log-probability-based TPO/PPO and also describes tokenized action likelihoods for both backbones. As written, the Pi0.5 path is still insufficiently specified and not yet internally coherent enough for reproducibility. I would need the exact objective, the log-probability definition used in PPO/TPO, the noise schedule/hyperparameters, and a clear reconciliation with the Appendix B description before considering this point resolved.

---

> > > ### Author Response · Authors · 2026-04-05
> > >
> > > Thank you for the continued engagement. We address each follow-up below.
> > >
> > > **1. PBRS invariance under POMDP.**
> > >
> > > We clarify that our paper does not claim policy invariance under reward shaping. We cite PRBS as the design principle motivating our potential-based form (Eq. 5), which ensures that the shaping signal redistributes reward across timesteps without introducing additional total reward.
> > >
> > > **2. Matched dense-nonstage baseline.**
> > >
> > > In STA-PPO, a naive "matched dense" baseline would be PPO with a single global distance reward (e.g., object-to-goal) at every step. However, stage decomposition and reward densification are inherently coupled by design: each stage defines a distinct potential function based on different geometric objectives. During Reach, the relevant distance is end-effector→object; during Transport, it is object→goal. A single global potential conflates these distinct objectives, providing misleading gradients.
> > >
> > > For STA-TPO (offline), isolation is feasible due to window-wise training. We note that the standard TPO baseline already uses fixed-window segmentation with sparse rewards. We provide ablations on ManiSkill3 (also presented to other reviewers previously):
> > >
> > > | Ablation | StackCube | LiftPegUpright |
> > > | :---- | :---- | :---- |
> > > | STA-TPO (stage-wise seg. \+ dense reward shaping) | 19.3% | 12.3% |
> > > | Random seg. \+ same dense shaping | 9.3% | 5.7% |
> > > | Random seg. \+ sparse reward | 7.3% | 4.7% |
> > > | Fixed-window seg. \+ same dense shaping | 16.3% | 8.0% |
> > > | TPO (Fixed-window seg. \+ sparse reward) | 15.7% | 7.7% |
> > >
> > > Dense shaping alone adds only +0.6% over TPO on StackCube (16.3% vs. 15.7%), while semantic decomposition adds a further +3.0%, confirming that structure—not density—drives the gains. Random segmentation actively hurts performance.
> > >
> > > **3. False-promotion scenario.**
> > >
> > > We directly address a specific case: if the robot is in the stage of Transport but then drops the object in the middle, the later-stage shaping reward is based on **object-to-goal** distance. Since the dropped object remains stationary, the potential $\Phi$ stays constant, producing a negative shaped reward $\gamma\Phi_{t+1} - \Phi_t = (\gamma-1)\Phi_t < 0$ at every subsequent step (task not completed), so the overall return signal is corrective. We agree that an explicit validity gate would further strengthen robustness for more complex tasks and will discuss this as a future direction.
> > >
> > > **4. Log-probabilities for $\pi_{0.5}$.**
> > >
> > > To apply PPO/TPO, we follow the Two-layer MDP formulation, which converts the deterministic ODE process into a stochastic SDE process by injecting Gaussian noise for stochasticity and tractable likelihood estimation.
> > >
> > > **(1). Stochastic Denoising Formulation:** The original flow update is reformulated into an SDE to enable stochasticity. At each denoising time $\tau \in [0,1]$, the state $\mathbf{x}_{\tau}$ is updated as:
> > >
> > > $$\mathbf{x}_ {\tau+\Delta\tau} = \mathbf{x}_ {\tau} + \left[\mathbf{v}_ {\theta}(\mathbf{x}_ {\tau},\tau) + \frac{\sigma_ {\tau}^{2}}{2\tau}\left(\mathbf{x}_ {\tau} + (1-\tau)\mathbf{v}_ {\theta}(\mathbf{x}_ {\tau},\tau)\right)\right]\Delta\tau + \sigma_ {\tau}\sqrt{\Delta\tau}\epsilon$$
> > >
> > > where $\epsilon \sim \mathcal{N}(0,\mathbf{I})$ and the noise schedule is
> > >
> > > $$\sigma_ {\tau} = a\sqrt{\frac{\tau}{1-\tau}}.$$
> > >
> > > Following the setup of $\pi_{\texttt{RL}}$, we set the noise level to $\mathbf{a=0.5}$ and use $\mathbf{N=4}$ denoising steps ($\Delta\tau = 1/N$). $\mathbf{N=4}$ was found to provide a sufficient balance between ODE-to-SDE discretization fidelity and computational efficiency.
> > >
> > > **(2). Log-probability Computation:** Under this formulation, each denoising step follows a Gaussian transition:
> > >
> > > $$\mathbf{x}_ {\tau_ {i+1}} \sim \mathcal{N}\left(\boldsymbol{\mu}_ {\theta}(\mathbf{x}_ {\tau_ i},\tau_ i), \sigma_ {\tau_ i}^{2}\Delta\tau \mathbf{I}\right).$$
> > >
> > > This yields a tractable step-wise log-probability:
> > >
> > > $$\log p(\mathbf{x}_ {\tau_ {i+1}} \mid \mathbf{x}_ {\tau_ i}) = -\frac{d}{2}\log(2\pi\sigma_ {\tau_ i}^{2}\Delta\tau) - \frac{\left\|\mathbf{x}_ {\tau_ {i+1}} - \boldsymbol{\mu}_ {\theta}(\mathbf{x}_ {\tau_ i},\tau_ i)\right\|^{2}}{2\sigma_ {\tau_ i}^{2}\Delta\tau},$$
> > >
> > > where $d$ is the action dimension. For PPO/TPO updates, the total surrogate policy log-probability is the sum over the denoising trajectory:
> > >
> > > $$\log \pi_ {\theta}(a \mid s) = \log p_ 0(\mathbf{x}_ 0) + \sum_ {i=0}^{N-1} \log p(\mathbf{x}_ {\tau_ {i+1}} \mid \mathbf{x}_ {\tau_ i})$$
> > >
> > > where $p_0(\mathbf{x}_0) = \mathcal{N}(0, \mathbf{I})$ is the initial noise distribution. Since $p_0$ is independent of $\theta$, it cancels in the PPO ratio and does not affect optimization.
> > >
> > > **(3). Implementation Summary:** By interpreting the denoising process as an **inner-loop MDP**, PPO/TPO is applied to these Gaussian transitions. This allows us to fine-tune flow-based VLAs using standard RL objectives. While OpenVLA uses tokenized action likelihoods, $\pi_{0.5}$ utilizes this denoising-trajectory surrogate likelihood.

---

### Official Review · Reviewer_oeDY · 2026-03-14

**Soundness:** 2
**Presentation:** 2
**Significance:** 3
**Originality:** 2
**Overall Recommendation:** 4
**Confidence:** 3

**Summary:**

This paper proposes STARE, a stage-aware reinforcement module for fine-tuning VLAs on long-horizon robotic manipulation tasks. The paper introduces STA-TPO for offline, stage-wise preference optimization, and STA-PPO for online, stage-aware reward shaping. The reported empirical results on SimplerEnv and ManiSkill3 are strong, with the method achieving very high success rates and showing especially large gains on precision-critical tasks.

**Compliance With Llm Reviewing Policy:**

Affirmed.

**Key Questions For Authors:**

- Are gains of the method over baselines due to stage-aware objectives, or partly due to the stronger multi-phase training recipe?
- Are there any failure cases for the proposed approach?

**Limitations:**

Please see weaknesses.

**Strengths And Weaknesses:**

## Strengths
- This paper is well-motivated and studies a relevant problem. The paper explains clearly why action reasoning differs from language reasoning and why stage-aware objectives are better suited to robotics.
- The proposed method is simple yet effective and is easy to follow.
- The reported gains are strong, showing high average success rates on SimplerEnv and ManiSkill3. It demonstrates particularly large improvements on tasks requiring precision and multi-stage coordination.

## Weaknesses
- STARE currently appears to rely on hand-designed stage definitions, thresholds, and geometric conditions. This raises several issues.
- The novelty of the proposed algorithm comes mainly from how individual building blocks are combined and adapted to VLA fine-tuning. That is still valuable, but the paper may overstate the algorithmic novelty unless it more clearly distinguishes what is genuinely new.
- The paper is primarily empirical and systems-oriented, which is fine. But some claims about better credit assignment and improved sample efficiency remain intuitive rather than formally justified.

---

> ### Author Rebuttal · Authors · 2026-03-31
>
> Thanks for recognizing our work. Below, we address each concern.
>
> ---
>
> **W1: Hand-designed stage definitions and thresholds.**
>
> STARE is based on rule-based geometric conditions. However, this is our deliberate design choice for *reliability and reproducibility*. Our stage definitions map directly to well-established manipulation primitives (reach distance, object height change, goal-distance potential...) that are **shared** across different manipulation tasks. Defining these requires only a few geometric thresholds per task family (not per task instance), and the same set of rules generalizes across all tasks within SimplerEnv and ManiSkill3, respectively.
>
> Importantly, as noted in Section 3.1 (Remark): "STARE is model-agnostic and can be naturally extended to learned event predictors or neural stage classifiers." Our rule-based design provides a strong, interpretable baseline that isolates the algorithmic contribution of stage-aware credit assignment from the noise introduced by learned decomposition. We will emphasize this point more clearly in the revision.
>
> **W2: Novelty.**
>
> We clarify the novelty of our contributions: STARE is based on existing components (PBRS, DPO/TPO, PPO). However, the novelty lies not in individual building blocks but in the insight that VLA fine-tuning requires *progressive* and *stage-aware* credit assignment—and in the concrete realization of this idea across both offline (STA-TPO, Eq. 6\) and online (STA-PPO, Eq. 5\) settings, and adapted them into a successful VLA RL fine-tuning pipeline. No prior VLA work has addressed these elements before.
>
> **W3: Credit assignment and sample efficiency lack formal justification.**
>
> More formal guarantees would strengthen the paper. However, we provide a detailed explanation of Algorithm 1 (STA-TPO) and Algorithm 2 (STA-PPO). We also provide substantial empirical evidence:
>
> - **CSSR analysis** (Figure 4): Directly shows STA-TPO improves conditional success at specific stages (e.g., Grasp \+27.2%, Place \+10.4%), demonstrating stage-level credit assignment.
>
> - **Stage-toggle ablation** (Figure 5): Removing STARE at precision-critical stages can cause \>20% drops, isolating the contribution of stage-aware signals.
>
> - **Same budget, higher performance** (Table 1,2): Under identical training steps and comparable wall-clock time, STARE methods achieved better results—demonstrating substantially more effective use of the same computational budget. We also provide computational overhead:
>
> | Component | Training Steps | Wall-clock |
> | :---- | :---- | :---- |
> | SFT | 100K | 23.2 hours |
> | TPO baseline | 50K | 23.8 hours |
> | STA-TPO (stage detection \+ shaping) | 50K | 24.5 hours (+3%) |
> | PPO baseline | 600K env steps | 49.0 hours |
> | STA-PPO (stage detection \+ shaping) | 600K env steps | 48.8 hours (≈0%) |
> | Full IPI | 100K \+ 50K \+ 600K env steps | 95.7 hours |
>
> *PutCarrotOnPlate task. A full breakdown will be provided in the revision.*
>
> **Q1: Are gains from stage-aware objectives or the multi-phase training recipe?**
>
> Table 1 isolates these contributions:
>
> - SFT→STA-TPO vs. GRAPE (same offline setting): Gains from stage-aware objectives alone.
>
> - SFT→STA-PPO vs. RL4VLA (same online RL setting): Gains from stage-aware rewards alone.
>
> - We also have IPI Ablation to isolate contributions cleanly:
>
> | Methods | Avg. Success |
> | :---- | :---- |
> | SFT → TPO → STA-PPO | 97.3% |
> | SFT → STA-TPO → PPO | 96.5% |
> | SFT → STA-TPO → STA-PPO (IPI) | 98.0% |
>
> This shows that both STA-TPO and STA-PPO contribute to the final performance for IPI. The full IPI pipeline (with STARE module) captures the cumulative benefit of both, with STA-PPO contributing the larger share of improvement. IPI training recipe pushes the SR to SOTA.
>
> This validates that both stage-aware components individually improve over their trajectory-level counterparts, and the entire pipeline represents cumulative improvements.
>
> **Q2: Failure cases.**
>
> Yes. We have identified three limitations of our work: (1) On simple tasks (e.g., PutCarrotOnPlate, PushCube), STA-PPO and PPO achieve similar final success rates. The improvement of STARE is most significant for problems where precision is important for each stage, such as Place in StackGreenOnYellow and Upright in LiftPegUpright (see Figure 3). (2) We have also tried insertion tasks, which have less structured manipulation semantics. It is very difficult to establish a clear stage decomposition for these problems. STARE provides limited improvement in our preliminary experiments. This highlights the dependence on well-defined stage decomposition. (3) In our real-world experiments, although STA-TPO outperforms the baselines, the quality of stage-wise preference data is limited due to inaccuracies in our perception pipeline (Grounding-DINO+STag), which sometimes resulted in poor-quality stage boundary annotations and reward signals. We have spent some time collecting good-quality preference data pairs.

---

> > ### Author Rebuttal · Reviewer_oeDY · 2026-04-04
> >
> > Thanks to the authors for the added experiments. After reading the rebuttal and other reviews, I decided to maintain my current score.

---

> > > ### Author Response · Authors · 2026-04-06
> > >
> > > We would like to thank Reviewer oeDY for their positive engagement. We are glad to see that the concerns have been fully resolved, and have now revised the manuscript with the clarification above.

---

### Decision · Program_Chairs · 2026-04-30

**Decision:**

Reject

**Comment:**

This paper received a mixed set of scores. Reviewers highlighted that the paper is well-motivated, well-written, and easy-to-follow. However, reviewers also pointed out that this paper depends on hand-designed stage designs and the formal justification for the stage-aware resward design. While many concerns have been addressed by authors' rebuttal, the main concerns of Reviewer rE8r still remains. While authors explained the paper did not claim invariance under reward shaping, the paper has a room for improving its clarity by clarifying the setup and assumption of the reward design, which I agree with and expect to need a major revision. Therefore, the paper cannot be accepted to the conference at this time. We encourage the authors to revise and resubmit to an appropriate future venue after incorporating the reviewers' comments.